# Parallel comparison of T cell and B cell subpopulations of adenoid hypertrophy and tonsil hypertrophy of children

Zihui Yu[1,6], Ziying Xu[1,6], Tongtong Fu[1,6], Shiyu Liu[1,2,6], Jinghua Cui[1], Bing Zhang[3], Jieqiong Liang[3], Chong Pang[3], Yuehua Ke[1], Ruikun Wang[1,4], Zhijie Tang[1,4], Yagang Gao[1], Bing Du[1], Yanling Feng[1], Hanqing Zhao[1], Guanhua Xue[1], Chao Yan[1], Lin Gan[1], Junxia Feng[1], Zheng Fan[1], Yang Yang[1], Lijuan Huang[1], Shuo Zhao[1], Sun Ying[5], Qinglong Gu[3] ✉ & Jing Yuan ◉[1] ✉

The adenoids and tonsils are important immune organs of the nasopharynx that often become hypertrophic in childhood because of recurrent pathogen infection. However, the differences in the immune microenvironment of adenoid hypertrophy (AH) and tonsil hypertrophy (TH) are unclear. Here, we show the epidemiological characteristics and peripheral blood cell indices of 1209 pediatric patients (1–15 years old) diagnosed with AH, and find that AH is often accompanied by TH and characterized by specific changes in immune cell types. Single-cell RNA sequencing analysis show that 12 paired AH and TH samples contain large numbers of B, T cells and some exhausted effector memory CD4+ T cells. Compared with matched TH, AH have more naïve B cells and regulatory CD4+ T cells and less plasma B cells. Weaker antigen presentation and more significant immunosuppression are also observed in AH. In contrast, the number and cytotoxicity of cytotoxic CD8+ T cells decrease with AH grade. These findings will help our understanding of the immune response to nasopharyngeal infection.

The adenoids (nasopharyngeal tonsils) and tonsils (palatine tonsils) are essential components of Waldeyer's lymphatic ring[1] and contain a diverse set of immune cells, such as T and B lymphocytes, plasma cells, macrophages, and dendritic cells. They serve as the front line against microbial invasion in childhood[2] and also play a role in preventing autoimmunity[3,4]. Numerous antigen-presenting cells, such as macrophages, dendritic cells, and helper T cells, are present in the crypts on the surface of the adenoids and tonsils, which, upon contact with pathogens, present exogenous antigens to T cells and B cells inside the gland[5]. After successful antigen recognition, helper T cells activate, proliferate, and differentiate into antigen-specific T cells. These T cells then activate naïve B cells, some of which differentiate into germinal center B cells, and then into plasma cells or memory B cells for antibody production[6]. Frequent viral and bacterial onslaughts can disrupt the balance between antigen invasion and immune clearance[7,8], leading to recurrent and chronic inflammation of the adenoids and/or tonsils, resulting in tissue hypertrophy[9].

Adenoid hypertrophy (AH) often occurs together with tonsil hypertrophy (TH)[10,11]. AH usually leads to upper airway obstruction with sleep-disordered breathing or even severe obstructive sleep apnea syndrome (OSAS). OSAS in children not only presents with symptoms of sleep distur- bances, but also with associated symptoms such as growth failure, neurocognitive and behav- ioral symptoms[12]. The symptoms of patients worsen when AH is accompanied by TH[13,14],

[1]Department of Bacteriology, Capital Institute of Pediatrics, Beijing 100020, China. [2]Military supplies and energy quality supervision station of Bejing, Beijing 100071, China. [3]Department of Otolaryngology, Capital Center For Children's Health, Capital Medical University, Beijing 100020, China. [4]Capital Institute of Pediatrics-Peking University Teaching Hospital, Beijing 100020, China. [5]School of Basic Medical Sciences, Capital Medical University, Beijing 100069, China. [6]These authors contributed equally: Zihui Yu, Ziying Xu, Tongtong Fu, Shiyu Liu. ✉e-mail: gql71@163.com; yuanjing6216@163.com

and aggravation of AH/TH can cause many other diseases to occur or worsen, such as pulmonary hypertension, insulin resistance, cognitive or neurocognitive deficits[15] and special facial changes[16].

In most adults, the adenoids appear in a residual form. The age-related morphological characteristics of tonsils are very similar to those of the adenoids, but the tonsils remain intact after puberty[17,18]. However, the immune defense capabilities of tonsils gradually decrease until they are lost. Specifically, the parenchymal area, lymphoid follicle area, and the proportion of germinal center B cells of tonsils decrease with age[19,20]. The different developmental trajectories of the adenoids and tonsils suggest that when faced with the same pathogen infection, they may exhibit different immune responses. Although a single-cell transcriptome profile of tonsils has been reported[21], no such profile has been reported for the adenoids. Furthermore, no parallel studies of the adenoids and tonsils of young children with disease states have been reported.

In this work, we analyze the epidemiology and peripheral blood cell index of 1209 pediatric patients (aged 1–15 years) diagnosed with AH. After characterizing the pathological features of AH as grade III–IV, single-cell RNA sequencing is performed on 12 paired AH and TH samples. Both AH and TH samples contain approximately 60% B cells, 30% T & natural killer (NK) cells, and effector memory CD4$^+$ T cells with an exhausted phenotype. Compared with matched TH, there is an increased proportion of naïve B cells and regulatory CD4$^+$ T cells, and a decreased proportion of plasma B cells in AH. In addition, compared with TH, AH has a stronger immunosuppressive ability, weaker antigen affinity, and a lack of interaction between B and T cell subpopulations. Our data augment current knowledge of the respiratory defense mechanism in early human development.

## Results

### Clinical characteristics of AH in pediatric patients

To clarify the epidemiological characteristics of AH, we collected the medical records of 1209 pediatric patients aged 1–15 years who had been diagnosed with AH at the Department of Otolaryngology, Capital Institute of Pediatrics, Beijing, between 2019 and 2021 (Fig. 1a). The Brodsky grading scale[22] and the Parikh classification[23] were used to determine tonsil and adenoid sizes, respectively. Among all patients, there were 78 cases of AH at grade I (6.45%), 81 cases at grade II (6.70%), 414 cases at grade III (34.24%), and 636 cases at grade IV (52.61%) (Table 1). Interestingly, as the grade of AH increased, the proportion of patients with TH grade IV decreased (Table 1). AH grade I was defined as the absence of hypertrophy (AH$^-$). The patients were mainly 2–5 years old (Fig. 1b). AH occurred in 1-year-old children, in whom the hypertrophy grade was the most serious. Furthermore, the age and body mass index of patients were negatively correlated with the AH grade (Fig. 1c, d, Supplementary Fig. 1a and Table 1), which may be related to enhancement of self resistance and physiological atrophy of the adenoids as the children get older, or to AH induced OSAS leading to growth restriction in children[12]. Analysis of the pathological features of AH grade III–IV showed that follicle size increased with hypertrophic grade and age up to 7 years, after which it gradually decreased. At the same age, follicle size in hypertrophic grade IV was larger than that in hypertrophic grade III (Fig. 1e and Supplementary Fig. 1b).

We then investigated changes in the immune cells of peripheral blood from these patients. There were upward trends in the numbers of white blood cells, monocytes, and lymphocytes in the peripheral blood of children with higher grade AH (Fig. 1f), but not in the ratios of lymphocytes, monocytes, neutrophils, and neutrophil number (Supplementary Fig. 1c). According to the presence of TH, these patients were divided into AH$^-$TH$^-$, AH$^+$TH$^-$, AH$^-$TH$^+$, and AH$^+$TH$^+$. The number of AH$^+$TH$^+$ patients was approximately 4.4 times that of AH$^+$TH$^-$ patients (921 vs 210) (Table 2). The age and body mass index of the AH$^+$TH$^+$ group were lower than those of the AH$^-$TH$^+$ group, but higher than those of the AH$^+$TH$^-$ group (Table 2 and Supplementary Fig. 1d). Compared with AH$^+$TH$^-$ patients, AH$^+$TH$^+$ patients had lower blood lymphocyte numbers and lymphocyte ratios (Supplementary Fig. 1e), but higher neutrophil numbers and neutrophil ratios (Supplementary Fig. 1f). In contrast, there was no significant difference in leukocyte number, monocyte number, or monocyte ratio between the two groups (Supplementary Fig. 1g). Notably, the same changes in lymphocyte number, lymphocyte ratio, neutrophil number, and neutrophil ratio between patients with TH and without TH were similarly observed in AH grade III and AH grade IV groups (Supplementary Fig. 1h–j). The levels of immune cells in the peripheral blood of patients with different AH grades and with/without TH showed various changes, indicating that AH and TH can activate a variety of immune cells. It was therefore important to more precisely characterize the immune microenvironment of AH and TH tissues.

### The immune microenvironment of AH in pediatric patients

To clarify the changes in the immune transcriptional status of the disease process, 24 samples (12 AH tissues and 12 matched TH tissues) were collected and processed from 12 patients (P01–P12) (Fig. 1a, Supplementary Fig. 2a, and Supplementary Table 1). None of the 12 patients had been exposed to cigarette smoke, and all had been immunized against pneumococci. Some patients (P01, P02, P07, and P11) had a history of allergy (Supplementary Fig. 2a and Supplementary Table 1). These patients were 2–5 years old, which was the high-incidence period of AH (Fig. 1a). Pathological diagnosis showed that all 12 patients had AH and TH, characterized by increased follicle size, inflammatory cell infiltration, and small vessel hyperplasia (Fig. 2a). In addition, follicle size increased with increasing hypertrophy grade (Supplementary Fig. 2b). Metagenomics next-generation sequencing (mNGS) was then performed on the 12 AH and 12 TH tissues to evaluate infection. The clustering tree results showed the correlation of gene abundance among 24 samples (Supplementary Fig. 2c). We found that the microbial composition of AH and TH was roughly the same, but AH contained a higher proportion of *Bacteroidetes* and *Deferribacteres* than TH (Supplementary Fig. 2d). At the species level, AH contained a higher proportion of *Desulfovibrio* and *Lachnospiraceae* than TH, while TH contained more *Haemophilus influenzae* than AH (Supplementary Fig. 2e). Moreover, linear discriminant analysis (LDA) effect size (LEfSe) difference analysis showed that AH contained more bacilli than TH, while TH contained more cocci than AH (Supplementary Fig. 2f).

Subsequently, separate analysis of scRNA-seq data from AH and TH groups revealed that both groups contained 7 major cell types, including B cells, T and NK cells, plasma B cells, myeloid cells, plasma cell-like dendritic cells, epithelial cells and neutrophils (Supplementary Fig. 3a–d). AH contained a higher proportion of B cells, myeloid cells and epithelial cells, while it contained a lower proportion of T and NK cells and plasma B cells (Supplementary Fig. 3e). Therefore, considering that the cell composition of the two tissues is very similar, we integrated the data of the two tissues into one dataset for subsequent analysis. Approximately 1 billion unique transcripts were obtained from 128,852 cells: 63,002 cells from adenoids and 65,850 cells from tonsils (Fig. 2b and Supplementary Fig. 3f-h). By graph-based uniform manifold approximation and projection (UMAP), seven high-confidence cell clusters were also identified, which could be assigned to known cell lineages (Fig. 2b). Seven major cell types were identified according to the expression of canonical gene markers (Fig. 2c and Supplementary Fig. 3i, j), including B cells (*CD79B$^+$*, *MS4A1$^+$*, *BANK1$^+$*), T and NK cells (*CD3D$^+$*, *CD3E$^+$*, *TRAC$^+$*), plasma B cells (*MZB1$^+$*, *XBP1$^+$*, *IGHG1$^+$*), myeloid cells (*LYZ$^+$*, *CST3$^+$*, *CD68$^+$*), plasma cell-like dendritic cells (pDC, *LILRA4$^+$*, *IL3RA$^+$*, *PLD4$^+$*), epithelial cells (*EPCAM$^+$*, *KRT18$^+$*, *PROM1$^+$*) and neutrophils (*FCGR3B$^+$*, *S100A9$^+$*, *CSF3R$^+$*).

AH and TH had similar cell composition, being mainly B cells and T&NK cells (Supplementary Fig. 4a). B cells accounted for 66.26% and 58.48% of the total number of cells in AH and TH, while T&NK cells

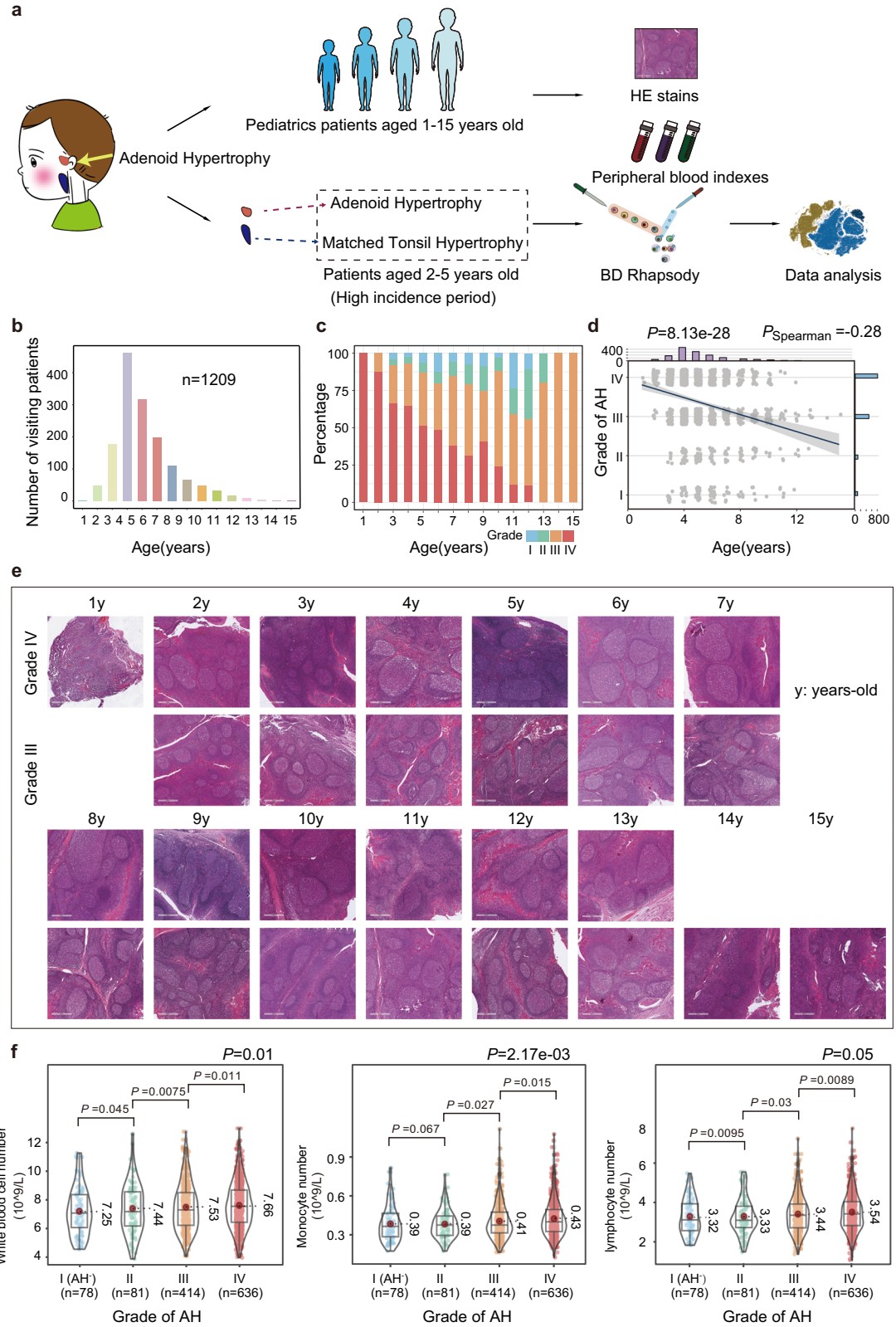

accounted for 29.47% and 35.61%, respectively. In addition, the proportions of plasma B cells in AH and TH were 2.75% and 4.75% (Supplementary Fig. 4a). For the 24 AH and TH samples, all these cell subtypes were present in all patients and in AH and TH, although in varying proportions (Supplementary Fig. 4b). Immunohistochemistry was conducted to provide an overview of the multicellular ecosystems of the AH and TH samples (Fig. 3c). Notably, the proportions of B cells

and myeloid cells increased, while those of T&NK cells, plasma B cells, and epithelial cells decreased in AH compared with TH (Fig. 2e and Supplementary Fig. 4c), which was consistent with the results of separate analysis of AH and TH data (Supplementary Fig. 3e). Moreover, these differences in cell proportion were not affected by the sex and allergy status of the patients (Supplementary Fig. 4d–g). We speculated that differences in the immune cell proportions between

**Fig. 1 | Overview of clinical information of children with AH. a** Schematic diagram of the study strategy. In order to clarify the epidemiological characteristics and immune microenvironment of AH, the clinical information, pathological sections and peripheral blood indexes were collected from 1209 pediatric patients (1-15 years old) diagnosed with AH from the Department of Otolaryngology at Capital Institute of Pediatrics from 2019 to 2021. Next, because of the high incidence of AH at the age of 2-5 years, 12 paired AH and TH samples were collected from same patients for scRNA-seq for identifying the similarities and differences between the immune microenvironment of AH and TH. **b** Histogram of age in children with AH. X-axis represents the age (years) of the children. Y-axis represents the number. **c** Bar plot illustrating the fraction of different AH grade at different ages (years). **d** Scatterplot showing the correlation between the age (years) in children with AH and the grade of AH. Spearman's rank correlation coefficient (Spearman's r) was used for evaluating correlation. The shaded area representing the 95% confidence interval. **e** Representative images of H&E-stained sections from pediatric patients (1-15 years old) diagnosed with grade III and IV of AH. Among them, 1 years-old patients only had grade IV of AH, while 14 and 15 years-old patients only had grade III of AH. Scale bar: 600 μm. **f** Violin and box plots showing the number of white blood cell number (left), monocyte number (middle) and lymphocyte number (right) from children with different grade of AH. Red dot represents the median value of each group. Data were expressed as median, upper and lower quartiles and 1.5× interquartile range in box-plots, and were expressed as maximum, minimum, and density distribution in violin-plots. *P* values were determined by a two-sided Welch's *t*-test and Welch's *F*-test. AH grade I was defined as AH⁻. Source data are provided as a Source data file.

**Table 1 | Demographic characteristics of different grade of AH**

|  | Grade of AH | | | |
|---|---|---|---|---|
|  | I (*n* = 78) | II (*n* = 81) | III (*n* = 414) | IV (*n* = 636) |
| **Sex** | | | | |
| Male sex[a] | 52 (66.67%) | 50 (61.73%) | 265 (64.01%) | 400 (62.89%) |
| Female sex[a] | 26 (33.33%) | 31 (38.27%) | 149 (35.99%) | 236 (37.11%) |
| **Age (years)** | 6.00 ± 2.05 (3–12) | 6.15 ± 2.37 (1–12) | 5.51 ± 2.18 (2–15) | 4.61 ± 1.65 (1–12) |
| **BMI (kg/m²)** | 17.54 ± 4.14 (10.33–35.71) | 18.39 ± 4.01 (13.23–28.91) | 16.87 ± 3.32 (11.98–30.47) | 16.16 ± 2.70 (12.11–26.45) |
| Underweight[b] | 6 (7.69%) | 3 (3.75%) | 46 (11.11%) | 95 (14.98%) |
| Healthy weight[b] | 44 (56.41%) | 36 (45.00%) | 236 (57.00%) | 389 (61.36%) |
| Overweight[b] | 9 (11.54%) | 12 (15.00%) | 48 (11.59%) | 49 (7.73%) |
| Obesity[b] | 19 (24.36%) | 29 (36.25%) | 84 (20.30%) | 101 (15.93%) |
| **Grade of TH** | | | | |
| NA[c]/I (TH⁻) | 5 (6.41%) | 10 (12.35%) | 60 (14.50%) | 130 (20.44%) |
| II | 0 (0.00%) | 0 (0.00%) | 4 (0.97%) | 6 (0.94%) |
| III | 54 (69.23%) | 56 (69.14%) | 278 (67.15%) | 414 (65.10%) |
| IV | 19 (24.36%) | 15 (18.51%) | 72 (17.38%) | 86 (13.52%) |

Data are *n* (%), or mean ± SD (min-max). All children included in this study were Chinese.
Source data are provided as a Source data file.
*AH* adenoid hypertrophy, *TH* tonsil hypertrophy.
[a]Male and female sex was referred to the biological sex of a baby at birth.
[b]The definition of BMI category was developed by the National Center for Health Statistics in collaboration with the National Center for Chronic Disease Prevention and Health Promotion (2000). http://www.cdc.gov/growthcharts. Three patients under the age of 2 were not included in the statistics.
[c]Nasopharyngeal endoscopy and CT have not yet detected TH.

AH and TH might be caused by different immune regulation and responses produced by adenoids and tonsils after infection.

Due to the absence of adenoids and tonsils in rats, mice, and hamsters[24], we identified the immune type changes in the nasopharyngeal lymph nodes of ovalbumin (OVA)-induced allergic rhinitis rat model[25] to determine the immune defense status of the nasopharynx. Peripheral blood serum and nasal lavage fluid were assayed for cytokines, and multiple interleukins and immunostimulatory factors were upregulated in the model group (Supplementary Fig. 4h). In addition, the follicles in the nasopharyngeal lymph nodes of the model group increased in size compared with those in the control group (Supplementary Fig. 4i). Moreover, the numbers of B cells, T&NK cells, myeloid cells, plasma cell-like dendritic cells, and neutrophils in the nasopharyngeal lymph nodes were higher in the model group (Supplementary Fig. 4j, k). These results indicate that the activated the type of immune cells during AH and TH are commonly present in the inflammatory response of the nasopharynx.

### Plasma B cells in AH show decreased proliferation and sensitivity to antigen stimulation

B cell numbers usually increase in infectious diseases and act to produce specific antibodies, mediate antigen presentation, release immune regulatory cytokines, and affect the functions of T cells and dendritic cells[26]. Our data showed that B cells were the most abundant cell type in AH and TH. Consistently, the B cell marker gene, CD20 (*MS4A1*), was highly expressed in the germinal center of follicles, indicating that extensive infiltration of B cells was a characteristic immune change in response to infectious disease (Fig. 3a). Moreover, the expression levels of CD20 in AH were higher than those in TH (Supplementary Fig. 5a). Next, we analyzed a total of 91,139 B cells and identified five subtypes by reclustering. These were naïve (Naïve B; *CD20⁺, CD27⁻, IGHD⁺*), atypical memory (atypical memory B; *CD20⁺, CD27⁻, DUSP4⁺, FCRL5⁺, ZEB2⁺, ITGAX⁺, FCRL4⁺⁺*)[27,28], germinal center (Germinal Center B; *CD20⁺, NEIL1⁺, RFTN1⁺, RGS13⁺*), proliferating (Proliferating B; *CD20⁺, MKI67⁺, STMN1⁺*) B cells and plasma B cells (*CD138⁺, IGHA1⁺, IGHG1⁺*) (Fig. 3b, c and Supplementary Fig. 5b–e).

Notably, compared with AH, there were some special proliferating B cells in TH, which highly expressed *IGHG1* (Fig. 3d, e), indicating that there were highly proliferating plasma B cells in TH. Proliferating B cells often exist in the dark areas of germinal centers where they undergo somatic hypermutation to obtain antibody affinity[29,30]. Using stacked plots and boxplots, the quantitative differences in B cell subtypes between the two tissues were identified. Compared with AH, the proportion of naïve B cells was significantly decreased, while the proportion of proliferating B cells and plasma B cells was significantly increased in TH (Fig. 3f). Similarly, these differences in cell proportion were not affected by the sex and allergy status of the patients (Supplementary Fig. 6a−d). Further Gene Ontology (GO) functional analysis

**Table 2 | Demographic characteristics of pediatric patients with AH**

| | AH⁻TH⁻ (n = 5) | AH⁺TH⁻ (n = 210) | AH⁻TH⁺ (n = 73) | AH⁺TH⁺ (n = 921) |
|---|---|---|---|---|
| **Sex** | | | | |
| Male sex[a] | 3 (60.00%) | 131 (62.38%) | 49 (67.12%) | 584 (63.41%) |
| Female sex[a] | 2 (40.00%) | 79 (37.62%) | 24 (32.88%) | 337 (36.59%) |
| **Age (years)** | 4.60 ± 1.20 (3–6) | 4.38 ± 1.83 (1–13) | 6.10 ± 2.06 (3–12) | 2.20 ± 1.99 (1–15) |
| **BMI (kg/m²)** | 17.51 ± 3.98 (13.45–16.23) | 15.88 ± 2.66 (12.37–28.91) | 17.72 ± 4.22 (10.33–35.71) | 16.73 ± 3.18 (11.98–30.47) |
| Underweight[b] | 0 (0.00%) | 31 (14.83%) | 6 (8.22%) | 113 (12.30%) |
| Healthy weight[b] | 5 (100.00%) | 142 (67.94%) | 39 (53.42%) | 519 (56.47%) |
| Overweight[b] | 0 (0.00%) | 12 (5.74%) | 9 (12.33%) | 97 (10.55%) |
| Obesity[b] | 0 (0.00%) | 24 (11.49%) | 19 (26.03%) | 190 (20.68%) |
| **Grade of AH** | | | | |
| I (AH⁻) | 5 (100.00%) | 0 (0.00%) | 73 (100.00%) | 0 (0.00%) |
| II | 0 (0.00%) | 10 (4.76%) | 0 (0.00%) | 71 (7.71%) |
| III | 0 (0.00%) | 63 (30.00%) | 0 (0.00%) | 351 (38.11%) |
| IV | 0 (0.00%) | 137 (65.24%) | 0 (0.00%) | 499 (54.18%) |
| **Grade of TH** | | | | |
| NA[c]/I (TH⁻) | 5 (100.00%) | 200 (95.24%) | 0 (0.00%) | 0 (0.00%) |
| II | 0 (0.00%) | 10 (4.76%) | 0 (0.00%) | 0 (0.00%) |
| III | 0 (0.00%) | 0 (0.00%) | 54 (73.97%) | 748 (81.22%) |
| IV | 0 (0.00%) | 0 (0.00%) | 19 (26.03%) | 173 (18.78%) |

Data are n (%), or mean ± SD (min-max). All children included in this study were Chinese.
Source data are provided as a Source data file.
[a]Male and female sex was referred to the biological sex of a baby at birth.
[b]The definition of BMI category was developed by the National Center for Health Statistics in collaboration with the National Center for Chronic Disease Prevention and Health Promotion (2000). http://www.cdc.gov/growthcharts. Three patients under the age of 2 were not included in the statistics.
[c]Nasopharyngeal endoscopy and CT have not yet detected TH.

showed that plasma B cells in AH highly expressed genes involved in immune-related pathways, such as humoral immunity and cell adhesion, indicating that plasma B cells in AH have a stronger ability to form humoral immunity than in TH (Fig. 3g). Plasma B cells in TH highly expressed genes involved in antigen-presentation pathways, such as in MHC and antigen processing, indicating that plasma B cells in TH might be more sensitive to antigen invasion (Fig. 3g). High expression of genes involved in MHC antigen recognition and other pathways was also observed in naïve B cells and proliferating B cells in TH (Supplementary Fig. 6e, f). However, naïve B cells in AH highly expressed interferon signaling pathway genes (Supplementary Fig. 6e); this expression is associated with specific differentiation and even exhaustion of naïve B cells[31]. These results indicate that B cells in AH might be more prone to exhaustion, while B cells in TH, especially plasma B cells, might be more sensitive to antigenic stimulation, and to also have high antibody affinity.

### AH and TH have differential proportions of naïve CD4⁺ T cells and regulatory CD4⁺ T cells

T&NK cells play an important role in anti-infection and bidirectional immune regulation[32,33]. To further reveal the functional differences between T cells and NK cells in AH and TH, we characterized them in the two tissues. Immunohistochemistry showed high levels of CD3 in both AH and TH samples, indicating extensive infiltration of T cells within the tissues (Fig. 4a). Moreover, the level of CD3 in AH was lower than that in TH (Supplementary Fig. 7a). Unsupervised clustering of T&NK cells showed five clusters of CD4⁺ T cells, two clusters of CD8⁺ T cells, and a population of double-negative T cells (DNT) and NK cells (Fig. 4c and Supplementary Fig. 7b–d). Furthermore, naïve (CD4-Naive; CD4⁺, CCR7⁺, SELL⁺, LEF1⁺), regulatory (CD4-Treg; FOXP3⁺, IL2RA⁺, IKZF2⁺), exhausted effector memory (CD4-Inhibitory1; ANXA1⁺, CD40LG⁺, TIGIT⁺), exhausted (CD4-Inhibitory2; PDCD1⁺, TIGIT⁺, CTL4⁺) and proliferating (CD4-Proliferation; MKI67⁺)

CD4⁺ T cells (Fig. 4c,d and Supplementary Fig. 7d–f) were also identified. Notably, unlike CD4-Inhibitory2 cells, which only highly expressed inhibitory markers (PDCD1, TIGIT, CTLA4), CD4-Inhibitory1 cells highly expressed both effector memory markers (S100A4, ANXA1, CD40LG) and inhibitory markers (PDCD1, TIGIT), indicating that the exhaustion of effector memory CD4⁺ T cells was obvious in AH and TH. Meanwhile, CD8-Naive cells with high expression of CD8A, LEF1 and TCF7 were defined as naïve CD8⁺ T cells. CD8-Cytotoxic cells corresponded to effector T cells because of the high expression of cytotoxic markers, such as NKG7, GZMA, and GZMK. Double-negative T cells were defined by their high expression of CD3G but lack CD4 and CD8A expression (Fig. 4c and Supplementary Fig. 7d–f). The accuracy of T cell subtypes was further clarified by gene set score, in which CD4-Naive and CD8-Naive T cells had the strongest immaturity, CD8-Cytotoxic and NK cells were the most cytotoxic, and CD4-Inhibitory1 and CD4-Inhibitory2 T cells had the strongest exhaustion state, although CD4-Inhibitory1 T cells also showed high effector memory function (Fig. 4d). Pseudotime trajectory analysis of CD4⁺ T cell subtypes after cell cycle correction revealed that naïve CD4⁺ T cells were the root, CD4-Proliferation cells were in the end state, while CD4-Inhibitory1, CD4-Inhibitory2 and CD4-Treg cells were in the intermediate state[33] (Fig. 4e and Supplementary Fig. 7g,h). Moreover, the pseudotime of CD4-Inhibitory1 cells was less than that of CD4-Inhibitory2 cells, indicating that the transition from CD4-Inhibitory1 to CD4-Inhibitory2 might be caused by the gradual loss of effector function of effector memory CD4⁺ cells and eventual complete exhaustion.

Comparison analysis of differences among T cell subtypes based on scRNA-seq showed that the relative percentage of naïve CD4⁺ T cells (CD4-Naive) decreased, while the percentage of regulatory CD4⁺ T cells (CD4-Treg) increased in AH compared with that in TH (Fig. 4f and Supplementary Fig. 8a–f). To further confirm these findings, we collected 9 additional pairs of AH and TH samples for flow cytometry

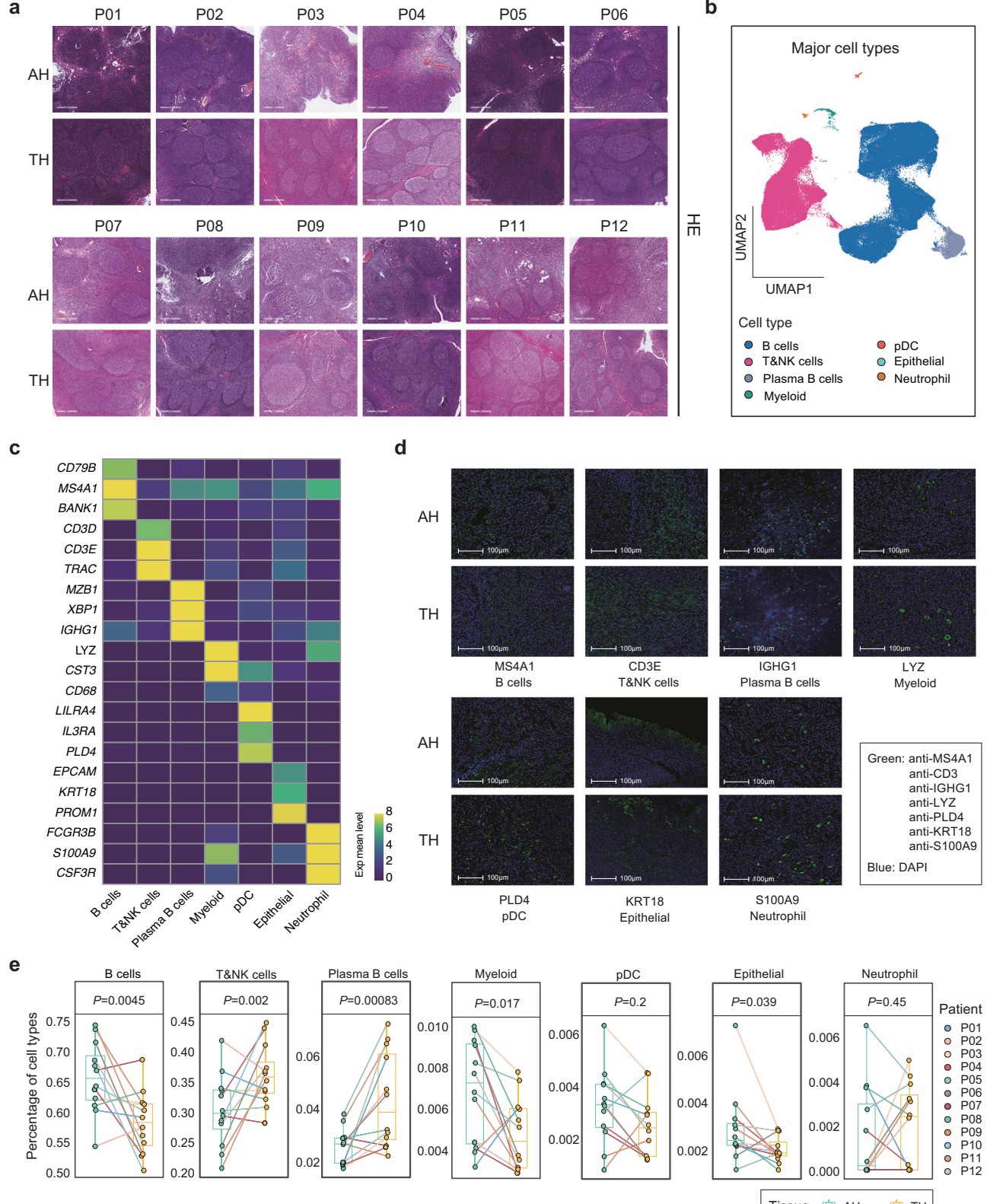

**Fig. 2 | scRNA-seq profiling of the AH and TH microenvironments.**
**a** Representative images of H&E-stained sections from 12 pairs of samples with AH and TH enrolled in the study. Scale bar: 600 µm. **b** UMAP of 128,852 cells post-QC and filtering grouped by major cell types. Each dot corresponds to a single cell, colored according to cell types. **c** Heatmap of canonical cell markers in major cell types. The color scheme is based on the average RNA expression distribution.

**d** Representative immunofluorescence images illustrating the canonical cell markers in major cell types from AH and TH. Scale bar: 100 µm. **e** Fractions of major cell types in each of the AH and TH samples ($n = 12$). Data were expressed as median, upper and lower quartiles and 1.5× interquartile range in box-plots. $P$ values were determined by a two-sided Wilcoxon's test.

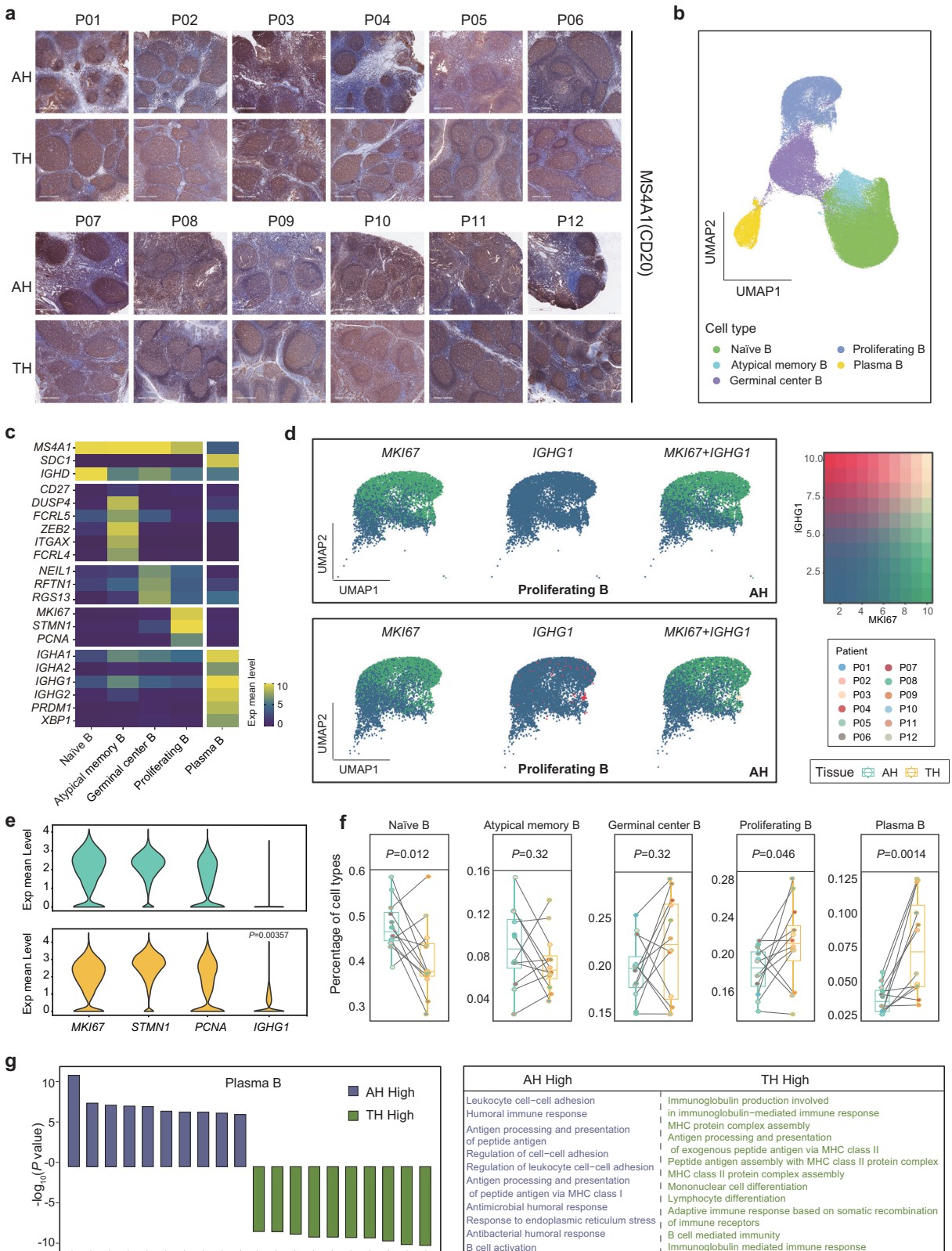

**Fig. 3 | scRNA-seq revealed heterogeneity in B cell subtypes in AH and TH.** **a** Representative IHC images of AH and TH tissue sections stained for CD20 from samples used for scRNA-seq. Scale bar: 600 μm. **b** UMAP plot showing the subtypes of B cells derived from AH and TH samples. Each cluster is color-coded according to cell types. **c** Heatmap of canonical cell markers in B cell subtypes. **d** UMAP plots showing the expression levels of *MKI67* and *IGHG1* in proliferating B cells of AH and TH. The color scheme is based on the co-expression correlation between two genes. **e** Violin plots showing the expression levels of *MKI67*, *STMN1*, *PCNA* and *IGHG1* in proliferating B cells of AH and TH. *P* values were determined by a two-sided Welch's *t*-test. **f** Fractions of B cell subtypes in each of the AH and TH samples (*n* = 12). Data were expressed as median, upper and lower quartiles and 1.5× inter-quartile range in box-plots. *P* values were determined by a two-sided Wilcoxon's test. **g** Enriched GO terms of differentially expressed genes obtained by comparing plasma B cells in AH to those in TH.

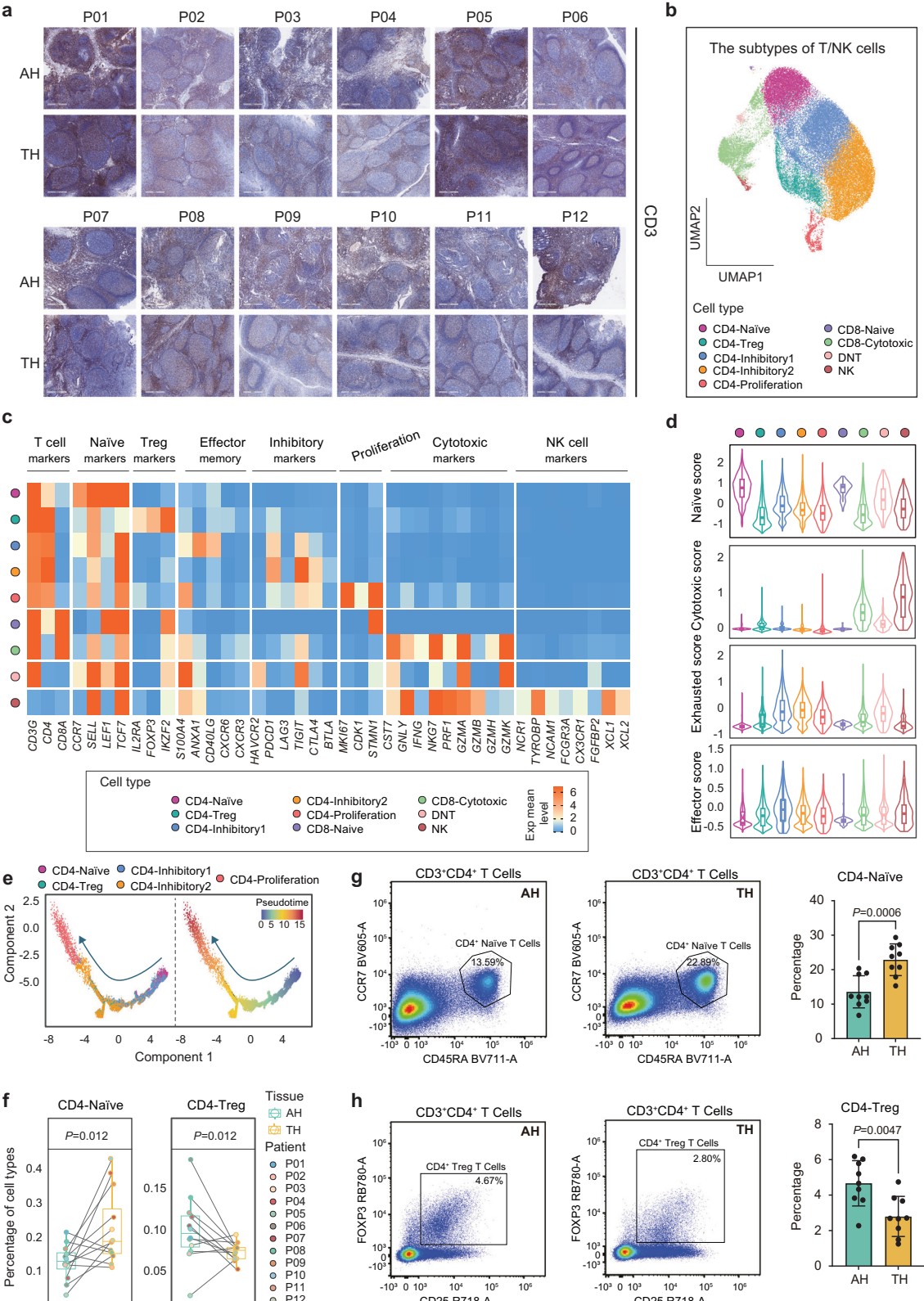

(Supplementary Table 2). The naïve CD4[+] T cells were selected by flow cytometry using antibodies against CD45, CD3, CD4, and CD45RA, while regulatory CD4 + T cells were selected by flow cytometry using antibodies against CD45, CD3, CD4, and CD25. The results showed that the proportion of naïve CD4[+] T cells in AH was lower than that in TH, while the proportion of regulatory CD4[+] T cells in AH was higher than that in TH (Fig. 4g, h).

## Functional analysis of B and T cells shows weaker antigen presentation and more significant immunosuppression in AH

We considered that the functional differences between B cells and T cells in AH and TH might indicate differences in their immune responses after adenoids and tonsils become hypertrophy. Therefore, to better understand the transcriptional heterogeneities of naïve CD4[+] T cells and regulatory CD4[+] T cells, substantial

**Fig. 4 | scRNA-seq revealed heterogeneity in T/NK cells in AH and TH.**
**a** Representative IHC images of AH and TH tissue sections stained for CD3 from samples used for scRNA-seq. Scale bar: 600 μm. **b** UMAP plot showing the subtypes of T/NK cells derived from AH and TH samples. Each cluster is color-coded according to cell types. **c** Heatmap of functional gene sets in T/NK subtypes. Treg, regulatory T cell. The color scheme is based on the average RNA expression distribution. **d** Violin and box plots showing the scores of naïve, cytotoxic, exhausted, and effector gene sets in each T/NK subtype. CD4-Naïve, $n = 8303$ single cells. CD4-Treg, $n = 3749$ single cells. CD4-Inhibitory1, $n = 13782$ single cells. CD4-Inhibitory2, $n = 10906$ single cells. CD4-Proliferation, $n = 776$ single cells. CD8-Naïve, $n = 57$ single cells. CD8-Cytotoxic, $n = 3755$ single cells. DNT, $n = 372$ single cells. NK, $n = 315$ single cells. Data were expressed as median, upper and lower quartiles and

1.5× interquartile range in box-plots, and were expressed as maximum, minimum, and density distribution in violin-plots. **e** Differential pseudotime trajectory analysis of CD4 T cell subtypes from AH and TH samples. CD4 T cell subgroups (left) and pseudotime (right) are labeled by colors. The arrows indicate the differentiation trajectory of the cells. **f** Fractions of naïve CD4$^+$ T cells and regulatory CD4$^+$ T cells in each of the AH and TH samples ($n = 12$). Data were expressed as median, upper and lower quartiles and 1.5× interquartile range in box-plots. $P$ values were determined by a two-sided Wilcoxon's test. Representative flow cytometry image and accumulated data showing the percentage of naïve CD4$^+$ T cells (**g**) and regulatory CD4$^+$ T cells (**h**) in AH and TH ($n = 9$ per group). Data are expressed as the mean ± SD in bar-plots. $P$ values were determined by a two-sided Welch's $t$-test. Source data are provided as a Source data file.

transcriptional program differences between 18 additional AH and TH samples (Supplementary Table 2) were defined using gene set variation analysis (GSVA) of hallmark pathways and flow cytometry analysis of immunofluorescence intensity. We found that naïve CD4$^+$ T cells of AH were enriched in pathways that support apoptosis and proliferation inhibition, while some T cell activation and differentiation pathways were highly expressed in this cell type of TH (Fig. 5a, c). In contrast, regulatory CD4$^+$ T cells of AH were enriched in T cell chemotaxis and cytokine production–related pathways, while regulatory CD4$^+$ T cells of TH were enriched in T cell tolerance and response to oxygen containing compounds-related pathways (Fig. 5b, d). These results indicate that that naïve CD4$^+$ T cells and regulatory CD4$^+$ T cells of AH and TH have different immune functions, respectively.

Analysis of scRNA-seq data to determine the proportion and function of B and T cell subtypes showed that compared with TH, B cells in AH exhibited stronger humoral reactions and exhaustion, as well as weaker proliferation, and antigen presentation. Meanwhile, T cells in AH were more likely to participate in immune suppression, apoptosis, and chemotaxis, but less likely to participate in proliferation, activation, and immune tolerance. To confirm these findings, we collected six pairs of additional AH and TH samples (Supplementary Table 3), and collected peripheral blood (PB) from children who came to the hospital for physical examination. Using antibodies against CD20 and CD3, B cells and T cells, respectively, were sorted from tissues and peripheral blood by flow cytometry. Compared with TH, AH contained a higher proportion of B cells and fewer T cells (Fig. 5e). After 36 hours of in vitro culture, cytokines were detected in the cell culture medium. We found that the levels of G-CSF, which promotes the production of B-cell activating factors[34], IL-8, MIP-1α, and MIP-1β, which increase B cell antigen affinity[35,36], IL-6 and MIF, which promote B cell maturation and proliferation[37,38], and SCGF-b,which reflects immune cell stemness), were lower in B cells of AH than in those of TH. Conversely, IL16, which inhibits plasma B cell production of IgE[39], was highly expressed in B cells of AH (Fig. 5f). It has been reported that G-CSF can promote immune suppression and apoptosis of T cells[40], IL-8, MIP-1α, and MIP-1β can promote T cell chemotaxis[41,42], IL-16 can inhibit T cell proliferation[43], MIF can promote adaptive immune response of T cells[44], while SCGF-β might reflect immune cell stemness. Their levels were higher in T cells of AH than in those of TH. The level of IL6, which promotes T cell activation[45]), was lower in T cells of AH (Fig. 5g). Notably, after lipopolysaccharide treatment, the levels of G-CSF, IL6, IL8, MIP1a, and MIP1b were upregulated in B and T cells of PB, AH and TH group, while the levels of IL16, MIF, and SCGFb were downregulated. But LPS did not alter the differential expression trend of these cytokines in B and T cells of AH and TH (Fig. 5f, g). However, compared with the PB group, LPS did not significantly increase the secretion of IL-8, MIP-1α, and MIP-1β, and IL-6 in B and T cells of AH and TH, which may be related to the already severe inflammatory response of B and T cells in AH and TH (Supplementary Fig. 9a,b). In addition, LPS had inhibitory effects

on the secretion of MIF, SCGF-β and IL-16 in PB, AH and TH (Supplementary Fig. 9a,b). These results indicate that the adenoids and tonsils, as well as the nasopharyngeal lymphatic system, have different immune response patterns after infection.

## Attenuation of cytotoxic CD8$^+$ T cells correlates with increased AH grade

Further analysis of functional gene expression by T cell subtypes revealed a significant negative correlation between the proportion of cytotoxic CD8$^+$ T cells and the grade of AH (Fig. 6a). Cytotoxic CD8$^+$ T cells play a key role in eliminating intracellular infection, and provide long-term protective immunity. However, in the context of disease, both intracellular and extracellular microenvironment changes can lead to dysfunction of cytotoxic CD8$^+$ T cells[46]. Therefore, we further examined whether there are any changes in the functions of cytotoxic CD8$^+$ T cells in AH. Analyses of cytotoxic score and the expression of *CST7*, *GZMA*, and *NKG7* indicated that the cytotoxicity of cytotoxic CD8$^+$ T cells also decreased with the progression of AH grade (Fig. 6b, c). To confirm these findings, we collected additional tissues from 18 AH stage III–IV cases, including nine in stage III and nine in stage IV (Supplementary Table 4) for analysis by flow cytometry and ELISA. The data showed that the number of cytotoxic CD8$^+$ T cells in stage IV was significantly reduced compared with that in stage III (Fig. 6d, e). Meanwhile, the level of granzyme A encoded by the cytotoxic gene *GZMA* decreased with the progression of AH (Fig. 6f).

Correspondingly, GSVA scores showed differences in the transcriptional function of cytotoxic CD8$^+$ T cells between the progression of AH and TH. Among them, the expression scores of the signaling pathways that mediate positive regulation of natural killer cell-mediated immunity and the positive regulation of tumor necrosis factor decreased with the progression of AH, while these trends were opposite in TH (Fig. 6g). Changes in the expression of *SLAMF6*, *HLA-F*, *CARD16*, *UBE2K*, *HSPA1A*, *HSPA1B*, *CASP4*, *CASP8*, and *OTULIN*, the core genes of the pathways, further confirmed the transcriptional differences in the progression of AH and TH (Fig. 6h). Similar to data shown in Fig. 6g, the pathways showing transcriptional change in AH and TH include the vitamin D biological process, the vitamin D metabolic process, and their core genes, *IFNG* and *CYP2R1*; that is, decreased expression with AH progression but increased expression with TH progression (Fig. 6g, h). Low serum vitamin D levels may be a risk factor for respiratory tract infections[47], and vitamin D deficiency can lead to impaired T cell proliferation and an imbalance in the proportion of cytotoxic T cells (CD8$^+$). Vitamin D supplementation can, however, restore the proliferation of T cells and normalize the proportion of T cell subtypes[46]. These findings indicate that decreases in the metabolism and use of vitamin D in cytotoxic CD8$^+$ T cells with the progression of AH grade may partly account for the decreasing cell number and cytotoxicity. However, changes in the regulation of tolerance induction, negative regulation of T cell-mediated cytotoxicity and in the expression of *PHLPP1*, *CD274*, *IL2RA*, *CEACAM1*, and *LILRB1* were opposite to the

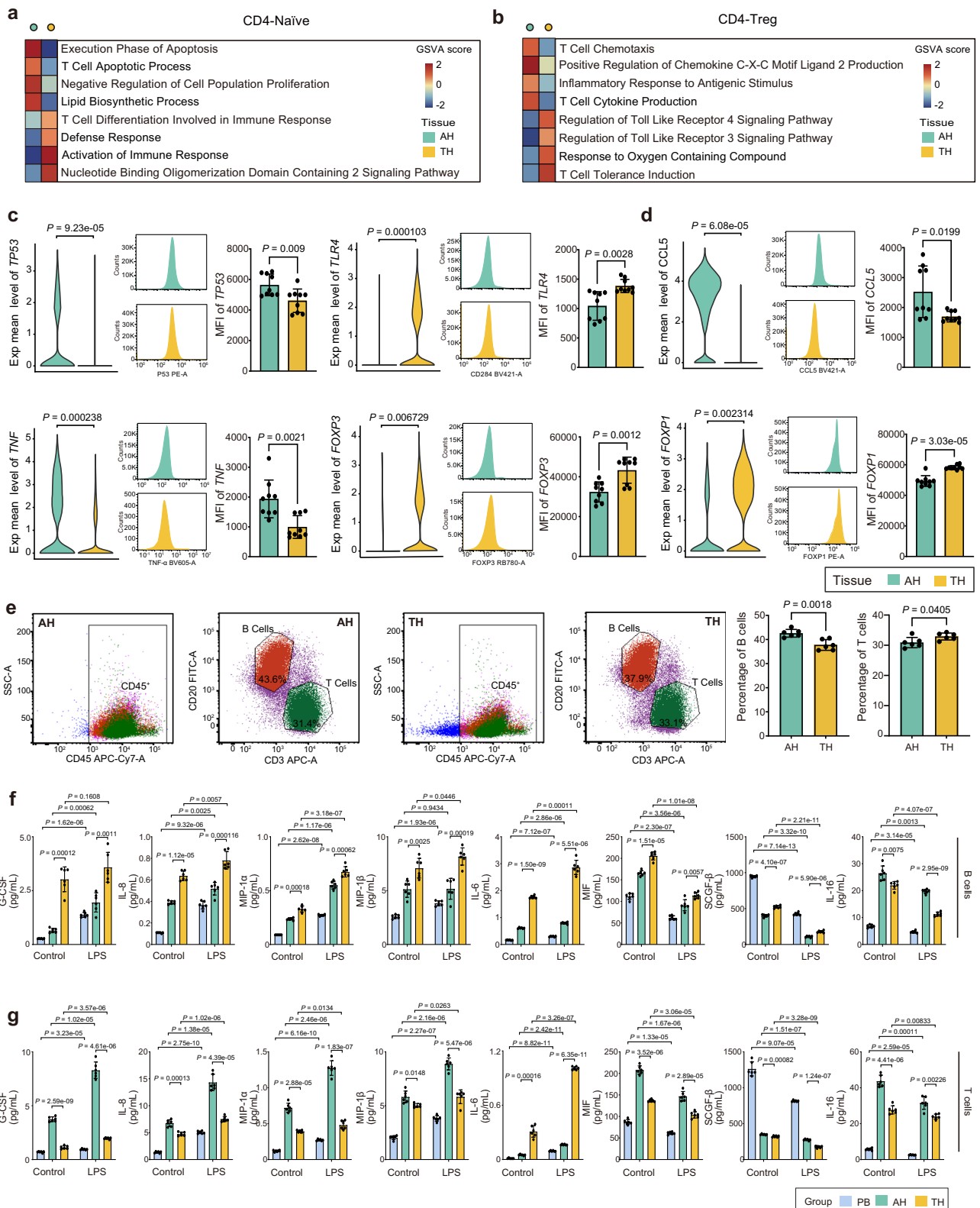

transcriptional changes of the above pathways and genes; again, the expression score increased with the progression of AH grade, and decreased with the progression of TH grade (Fig. 6g, h). The immune tolerance of CD8[+] T cells in chronic hepatitis B might be a reason why the hepatitis B virus is not effectively eliminated[48]. Taken together, the decrease in the number of cytotoxic CD8[+] T cells and the attenuation of cytotoxicity may be an immune mechanism that drives the progression of AH grade.

**Cross talk is reduced between exhausted CD4[+] T cells and plasma B cells and between NK cells and naïve B cells in AH**

Crosstalk between T and B cells often plays an important role in immune responses. We therefore analyzed cell communication by characterizing the expression of ligand-receptor pairs in T and B cell subtypes. The results showed increases cell communication between CD4-Inhibitory and CD4-Proliferation to plasma B, and increased communication between NK cells and Naïve B, memory B, GC, and

**Fig. 5 | Functional differences between B cells and T cells in AH and TH. a, b** Heatmap showing differences in pathways scored per cell by GSVA between T/NK subtypes of AH and TH. Differential pathways of naïve CD4⁺ T cells are on the left panel (**a**), while the differential pathways of regulatory CD4⁺ T cells are on the right panel (**b**). **c** The expression levels of *TP53*, and *TLR4* in naïve CD4⁺ T cells of AH and TH. Violin plots showing the expression levels of *TP53*, and *TLR4* in naïve CD4⁺ T cells of AH and TH through the analysis of scRNA-seq (AH, $n = 2553$ single cells; TH, $n = 5750$ single cells). Data were expressed as maximum, minimum, and density distribution (left), *P* values were determined by a two-sided Welch's *t*-test and adjusted for multiple comparisons. Representative results of fluorescence intensity and cell counts of P53 (*TP53*), and CD284 (*TLR4*) in naïve CD4⁺ T cells of AH and TH analyzed by flow cytometry (middle). The mean fluorescence intensity (MFI) of P53 (*TP53*), and CD284 (*TLR4*) in naïve CD4⁺ T cells of AH and TH analyzed by flow cytometry. Data were expressed as the mean ± SD (right) ($n = 9$ per group), *P* values were determined by a two-sided Welch's *t*-test. *TP53* is a core gene involved in execution phase of apoptosis, T cell apoptotic process, and negative regulation of cell population proliferation pathways. *TLR4* is a core gene involved in nucleotide binding oligomerization domain, activation of immune response, and defense response pathways. **d** The expression levels of *CCL5*, *TNF*, *FOXP3*, and *FOXP1* in regulatory CD4 + T cells of AH and TH. Violin plots showing the expression levels of *CCL5*, *TNF*, *FOXP3*, and *FOXP1* in regulatory CD4⁺ T cells of AH and TH through the

analysis of scRNA-seq (AH, $n = 1704$ single cells; TH, $n = 2045$ single cells). Data were expressed as maximum, minimum, and density distribution (left), *P* values were determined by a two-sided Welch's *t*-test and adjusted for multiple comparisons. Representative results of fluorescence intensity and cell counts of CCL5, TNF, FOXP3, and FOXP1 in regulatory CD4⁺ T cells of AH and TH analyzed by flow cytometry (middle). The mean fluorescence intensity (MFI) of CCL5, TNF, FOXP3, and FOXP1 in regulatory CD4⁺ T cells of AH and TH analyzed by flow cytometry. Data were expressed as the mean ± SD (right) ($n = 9$ per group), *P* values were determined by a two-sided Welch's *t*-test. *CCL5* is a core gene involved in T cell chemotaxis pathways. *TNF* is a core gene involved in positive regulation of chemokine C-X-C motif ligand2 production and inflammatory response to antigenic stimulus pathways. *FOXP3* is a core gene involved in T cell tolerance pathways. *FOXP1* is a core gene involved in response to oxygen containing compound pathways. **e** Representative flow cytometry image and accumulated data showing the percentage of B cells and T cells in AH and TH ($n = 6$ per group). Data were expressed as the mean ± SD. *P* values were determined by a two-sided Welch's *t*-test. The conce($n = 6$ per group)ntration of G-CSF, IL-8, MIP-1α, MIP-1β, IL-6, MIF, SCGF-β, IL-16 secreted by B (**f**) and T (**g**) cells in peripheral blood of children without inflammation, AH, and TH. PB, peripheral blood. Data were expressed as the mean ± SD. *P* values were determined by a two-sided Welch's *t*-test. Source data are provided as a Source data file.

proliferating B cells in TH compared with that in AH (Fig. 7a and Supplementary Fig. 10a). To elucidate the specific cell communication process, ligand-receptor pairs highly expressed by these cells were identified in TH (Fig. 7b, c). It is reasonable to speculate that the correlation between the proportion of different cell types can be used to judge the recruitment/inhibition effect between the cells. Furthermore, it was noted that there were positive correlations between the proportion of inhibitory CD4⁺ T cells and plasma B cells, and between NK cells and memory B cells, while there was a negative correlation between the proportion of NK cells and naïve B cells in TH (Fig. 7d and Supplementary Fig. 10b), but not in AH (Fig. 7d).

To further identify the ligand–receptor pairs involved in recruitment/inhibition between cell types, we generated a scatter plot of ligand gene expression and the number of cell types recruited/inhibited. The results showed that in the ligand-receptor pair *PTPRC/CD22* of TH, there was a positive correlation between the abundance of plasma B cell- and inhibitory CD4⁺ T-cell-derived *PTPRC* expression (Fig. 7e). Mechanistically, multiplex immunofluorescence staining revealed that, compared with AH, there was increased expression of *PTPRC/CD22* in inhibitory CD4⁺ T cells and plasma B cells in TH (Fig. 7f and Supplementary Fig. 10c,d). Similarly, there was a negative correlation between the abundance of naïve B cells and NK-cell-derived *CCL4* expression in TH (Fig. 7g). Immunofluorescence further demonstrated that NK cells and naïve B cells increased the expression of *CCL4/CNR2* in TH (Fig. 7h and Supplementary Fig. 10c,d). However, ligand–receptor pairs that exert recruitment effects were not observed, either in NK cells or in memory B cells (Supplementary Fig. 10e). These results indicate that the abundant crosstalk between T and B cells in TH might form a unique immune microenvironment compared with AH, providing a reason for the differences in the number of plasma B cells and naïve B cells between the two tissues. In addition, the epithelial cells were important factors in triggering immune regulation. We found through further cell communication analysis that compared to TH, increases cell communication between the epithelial cells and naïve CD4⁺T cells, regulatory CD4⁺ T cells, exhausted effector memory CD4⁺ T cells, exhausted CD4⁺ T cells, cytotoxic CD8⁺ T cells and DNT in AH (Supplementary Fig. 11a), which might be due to the role played by ligand-receptor pairs *LGALS9/CD45* (Supplementary Fig. 11b). These results indicate the presence of a unique immune regulatory pattern of epithelial cell activated T cell subtypes in AH.

## Discussion

The adenoids and tonsils degenerate with age, but this occurs earlier in the adenoids[17,49,50]. This suggests that the two tissues have different

immune defense mechanisms. The present study comprehensively decoded the transcriptional changes of regulatory CD4⁺ T cells, naïve B cells, and plasma B cells in AH. We revealed that a unique immune regulatory pattern of epithelial cell activated T cell subtypes in AH, while B cells and T cells in AH were more inclined to exhaustion and immunosuppression, but less prone to antigen presentation, proliferation, and activation compared with those in TH, indicating that the two tissues have unique immune response patterns and play the main immune function through B cells and T cells. Furthermore, the specific characteristics of cytotoxic CD8⁺ T cells that are associated with particular grades of AH were also identified. Our parallel profiling of the immune microenvironment of AH and matched TH at the single-cell level revealed immune response differences between the adenoids and tonsils after hypertrophy by infection, and enabled us to document the molecular characteristics of B cell and T cell subpopulations in the diseases at a high resolution.

Specifically, compared to TH, AH contained more B cells, especially naïve B cells, and the interferon production function of naïve B cells was also activated. Normally, interferon signaling in B cells is rapidly activated in response to pathogenic microorganism infection[51]. Therefore, it is reasonable to speculate that naïve B cells in the adenoids are more sensitive to pathogenic microorganism infection. However, interferon signaling can drive the specific differentiation of naïve B cells, leading to B cell exhaustion[31]. After B cells leave the bone marrow, they gradually mature with help from CD4⁺ T cells and differentiate into plasma B cells or memory B cells, which produce large amounts of antibodies[52]. Notably, our data indicate that some plasma B cells in TH may have increased antibody affinity and antigen recognition ability. For T cells, AH contained more regulatory CD4⁺ T cells, which exhibited more significant chemotaxis and weaker tolerance. Regulatory CD4⁺ T cells play an immunosuppressive role in the adenoids and promote persistent pneumococcal infection in children[53], which might represent a unique pattern of regulatory CD4⁺ T cells driving the progression of AH. Meanwhile, there were fewer naïve CD4 + T cells in AH, while CD4⁺ T cell activation of naïve CD4⁺ T cells require certain signals to proliferate and acquire functional differentiation[54]. Furthermore, the functional analysis showed that compared to TH, naïve CD4⁺ T cells of AH lacked the ability to regulate T cell proliferation and activation[55]. Meanwhile, by evaluating the transcriptome and cytokine secretion of B and T cells in AH and TH, we found that, compared with TH, the formation of B-cell-based humoral immunity and T cell-mediated cell chemotaxis might be more rapid in AH, while the ability to activate immune cell proliferation and immune tolerance might be relatively weak, meaning that immune suppression and apoptosis were more likely to occur. These results

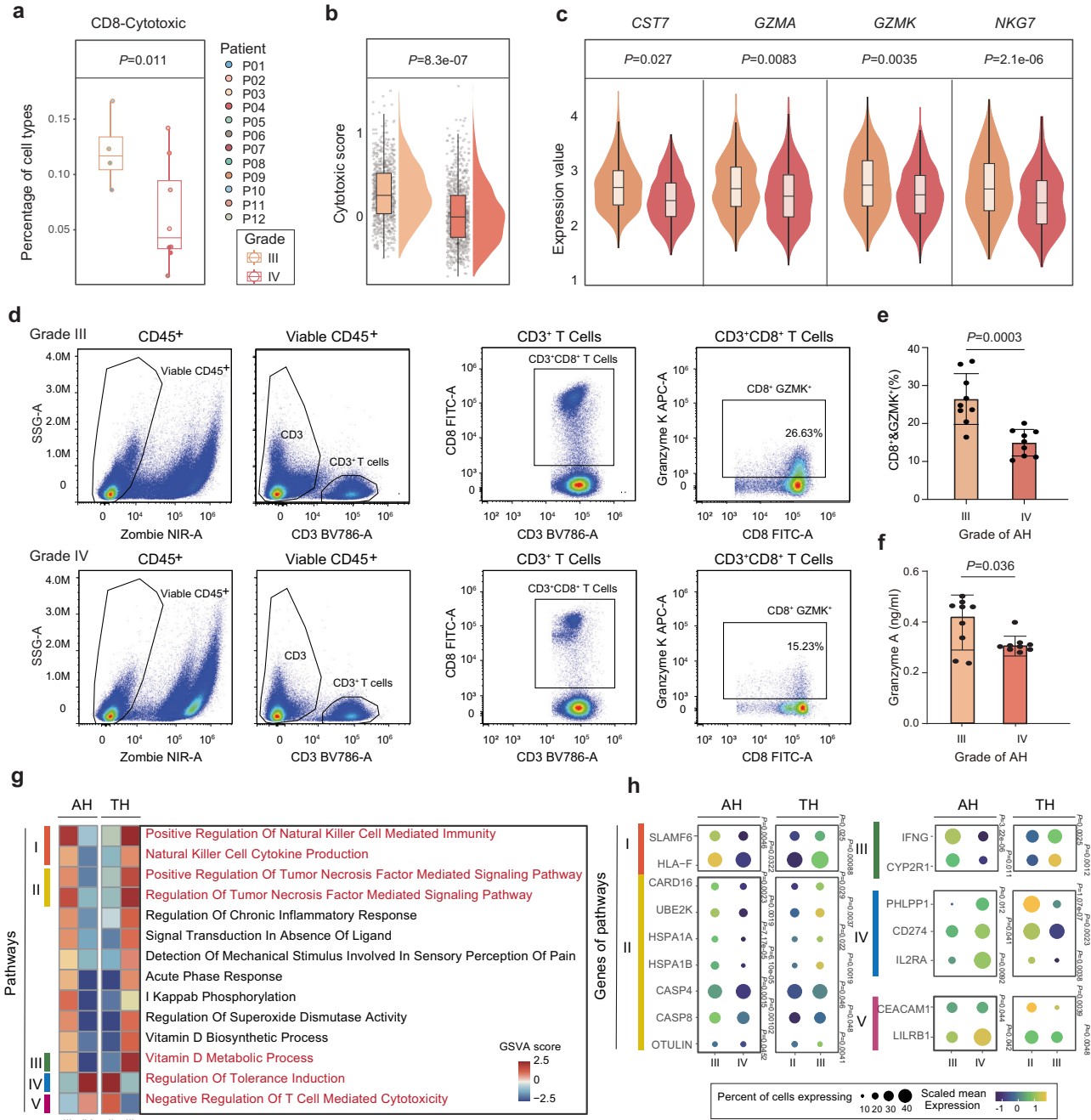

**Fig. 6 | Cytotoxic CD8+ T cells in the progression of AH. a** Fractions of cytotoxic CD8+ T cells in each of different grade of AH samples (Grade III, $n = 4$; Grade IV, $n = 8$). Data were expressed as median, upper and lower quartiles and 1.5× interquartile range in box-plots. $P$ values were determined by a two-sided Wilcoxon's test. **b** Violin and box plots showing the scores of cytotoxic gene sets in different grade of AH (Grade III, $n = 436$ single cells; Grade IV, $n = 1281$ single cells). Data were expressed as median, upper and lower quartiles and 1.5× interquartile range in box-plots, and were expressed as maximum, minimum, and density distribution in violin-plots. $P$ values were determined by a two-sided Welch's t-test. **c** Violin and box plots showing the expression level of *CST7, GZMA, GZMK* and *NKG7* in different grade of AH (Grade III, $n = 436$ single cells; Grade IV, $n = 1281$ single cells). Data were expressed as median, upper and lower quartiles and 1.5× interquartile range in box-plots, and were expressed as maximum, minimum, and density distribution in

violin-plots. $P$ values were determined by a two-sided Welch's t-test. Representative flow cytometry image (**d**) and accumulated data (**e**) showing the percentage of cytotoxic CD8+ T cells in different grade of AH ($n = 9$ per group). Data were expressed as the mean ± SD. $P$ values were determined by a two-sided Welch's t-test. **f** Expression level of granzyme A in different grade of AH ($n = 9$ per group). Data were expressed as the mean ± SD. $P$ values were determined by a two-sided Welch's t-test. **g** Heatmap showing differences in pathways scored per cell by GSVA between different grade of AH and TH. The color scheme is based on the GSVA score. **h** Dot plot showing the expression level in genes of the pathways in Fig. 2g from cytotoxic CD8+ T cells in different grade of AH and TH. $P$ values were determined by a two-sided Wilcoxon's test and adjusted for multiple comparisons. The color scheme is based on the scaled average RNA expression. Source data are provided as a Source data file.

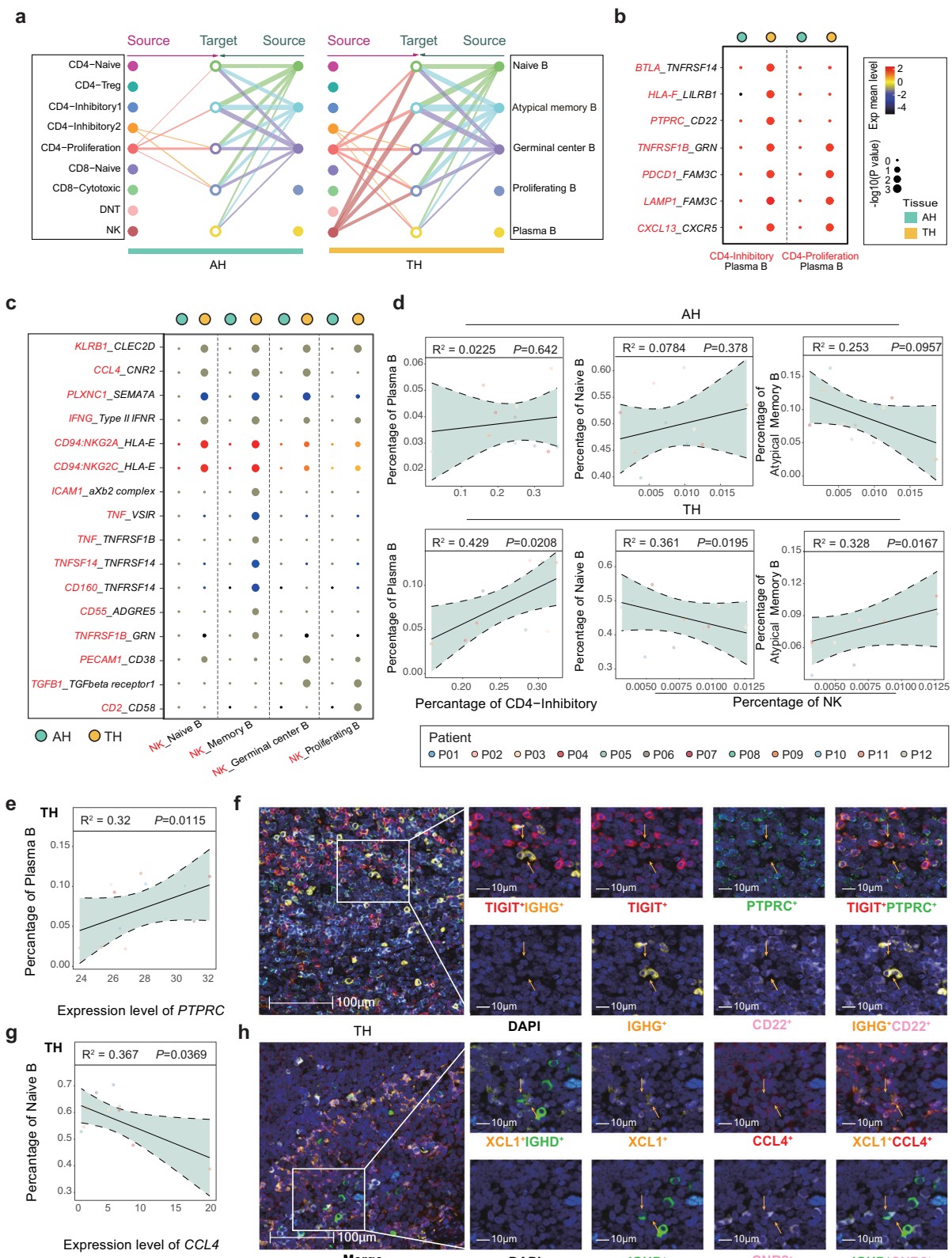

revealed that although the adenoids and tonsils both act as immune organs in the nasopharynx, they have different immune defense functions. Interestingly, TH has more abundant interactions between subsets of T and B cells, which further supports the ability of tonsils to maintain an immune response.

The effector memory CD4$^+$ T cells have the potential to rapidly secrete cytokines and promote the memory response of tissues to reinfection[56]. Our results show that effector memory CD4$^+$ T cells were exhausted in both AH and TH, indicating that under repeated exposure to infective pathogens, the responses of memory CD4$^+$ T cells in the adenoids and tonsils decrease, which might account for disease progression. In addition to CD4$^+$ T cells, cytotoxic CD8$^+$ T cells also play an important roles in anti-infection immunity and in clearing tumor cells[57–59]. In a variety of tumors, cytotoxic CD8$^+$ T cells gradually lose their effector function, but express more inhibitory receptors, and enter an exhausted state because of sustained antigen stimulation and

**Fig. 7 | Interactions between T/NK cell subtypes and B cell subtypes in AH and TH. a** Hierarchical plot showing the intercellular interactions among different cell types in AH (left) and TH (right). Solid and open circles represent source and target, respectively. Edge width represents the communication probability. Edge colors are consistent with the signaling source. **b** Dot plot showing intercellular interactions between exhausted CD4[+] T cells and plasma B cells, proliferating CD4[+] T cells and plasma B cells. Circle size indicates the significance of the interaction, and circle color indicates the mean expression of ligand and receptor genes. The red letters represent ligands, and the black letters represent receptors. The color scheme is based on the mean expression distribution. *P* values were determined by a two-sided Welch's *t*-test and adjusted for multiple comparisons. **c** Dot plot showing intercellular interactions between NK cells and naïve B cells, atypical memory B cells, germinal centre B cells, proliferating B cells. Circle size indicates the significance of the interaction, and circle color indicates the mean expression of ligand and receptor genes. The red letters represent ligands, and the black letters represent receptors. The color scheme is based on the mean expression distribution. *P* values were determined by a two-sided Welch's *t*-test and adjusted for multiple comparisons. **d** Scatterplot showing the correlation between the relative ratio of exhausted CD4[+] T cells and plasma B cells, NK cells and naïve B cells, NK cells and atypical memory B cells from AH (top) and TH (bottom) in the scRNA-seq dataset. Pearson's correlation coefficient (Pearson's r) was used to evaluate correlation. The shaded area representing the 95% confidence interval. *P* values were

determined by Pearson-correlation test. **e** Scatterplot showing the correlation between the expression of PTPRC in exhausted CD4[+] T cells and the proportion of plasma B cells from TH in the scRNA-seq dataset. Pearson's correlation coefficient (Pearson's r) was used to evaluate correlation. Spearman-correlation test. The shaded area representing the 95% confidence interval. *P* values were determined by Pearson-correlation test. **f** Representative immunofluorescence images illustrating the interaction between exhausted CD4[+] T (TIGIT[+]) cells and plasma B cells (IGHG1[+]) through PTPRC/CD22 in one TH sample (P06). The small panels show the magnification of the selected region highlighted in red. Yellow arrows indicate the colocalization of exhausted CD4[+] T cells and plasma B cells. Scale bars correspond to 100 μm and 10 μm in the large and small panels, respectively. **g** Scatterplot showing the correlation between the expression of CCL4 in NK cells and the proportion of naïve B cells from TH in the scRNA-seq dataset. Pearson's correlation coefficient (Pearson's r) was used to evaluate correlation. The shaded area representing the 95% confidence interval. *P* values were determined by Pearson-correlation test. **h** Representative immunofluorescence images illustrating the interaction between NK cells (XCL1[+]) and naïve B cells (IGHD[+]) through CCL4/CNR2 in one TH sample (P2). The small panels show the magnification of the selected region highlighted in red. Yellow arrows indicate the colocalization of NK cells and naïve B cells. Scale bars correspond to 100 μm and 10 μm in the large and small panels, respectively.

---

an immunosuppressive microenvironment[60–63]. However, we did not observe exhausted cytotoxic CD8[+] T cells, either in AH or in TH. Interestingly, as AH progressed, the number of cytotoxic CD8[+] T cells gradually decreased, and the cytotoxicity gradually weakened. The expression of cytotoxic genes *CST7*, *GZMA*, and *NKG7* was consistently downregulated, which possibly also drives AH progression.

Our study has several limitations. First, we were unable to analyze changes in cell types other than T and B cells because of the very small number of cells captured, especially epithelial cells. Epithelial cells were important factors in triggering immune regulation, therefore, in future research, sample preparation methods should be optimized, with a focus on examining the differences in epithelial cells between AH and TH. Second, the clonal relationship of T and B cells was not accurately identified. Therefore, future studies should use technologies with higher recognition, such as spatial transcriptomics and immune profiling of T and B cell receptors, which may further clarify the immune microenvironment of AH and TH.

In summary, we have comprehensively analyzed the immune cell states of AH and paired TH, and have detailed the difference in the immune response between the adenoids and tonsils. Our findings provide an additional perspective for understanding how the respiratory system defends against infection.

## Methods

### Clinical information and sample collection

All samples were obtained from Department of Otolaryngology, Capital Institute of Pediatrics, Beijing, China. Pathological sections borrowed from the Department of Pathology, Capital Institute of Pediatrics, Beijing, China. This study was approved by the Ethics Committee Board of Capital Institute of Pediatrics with number SHERLL2021077, and the written informed consent was obtained from all participants included in this study. All participants provided samples free of charge. The animal experiments were approved by the Research Ethics Committee of Institute of Chinese Materia Medica, China Academy of Chinese Medical Sciences (2024B162) and complied with ethical standards and international conventions on animal experimentation.

### Complete blood count measurement

The blood samples were drawn of 1209 patients in the EDTA tubes for measurement of the complete blood cell counts. We calculated the indexes of white blood cell number, monocyte number, lymphocyte number, neutrophils number, lymphocytes ratio, monocytes ratio, and neutrophils ratio. "The number" refers to the number of cell type in 1 L

peripheral blood. "The ratio" refers to the proportion of cell type in the total number of white blood cells in 1 L peripheral blood. Cell counting was analyzed using Flow Cytometry method with semiconductor laser on Automated hematology analyzer Sysmex.

### Single-cell suspension preparation

The collected fresh tissues were cut into small pieces, ground with the plunger of a plastic syringe at stainless steel screen (250 μm), and then cleaned with Hanks buffer. The grinding and cleaning were repeated several times to release as many cells as possible from the interstitium. After collecting, the cells were suspended by repeated pipetting, using 40 μm cell sieve and placed on ice for standby. Finally, 24 samples of single-cell suspension were obtained.

### BD Rhapsody single-cell RNA-seq

The library was constructed using the BD Rhapsody Single-Cell Analysis System (BD, Biosciences). Cells from each sample were labeled with sample tags (BD Human Single-Cell Multiplexing Kit) following the manufacturer's protocol. Briefly, a total number of $5 \times 10^6$ cells was resuspended in 180 mL of Stain Buffer (FBS) (BD PharMingen). The sample tags were added to the respective samples and incubated for 20 min at room temperature, every 4 samples (including 2 adenoid tissues and matched 2 tonsil tissues) were mixed in the order of P1-P12 using BD Human Single-Cell

Multiplexing Kit (BD, 633781). After using this kit to process samples, we could combine and process different samples simultaneously to reduce costs, but in subsequent data analysis, the sample labels were automatically de-multiplexed and individual samples could be identified. For each pooled sample two BD Rhapsody cartridges were super-loaded with approximately 60,000 cells each. Single cells were isolated using Single-Cell Capture and cDNA Synthesis with the BD Rhapsody Express Single-Cell Analysis System according to the manufacturer's recommendations (BD Biosciences). The cDNA libraries were prepared using the BD Rhapsody Whole Transcriptome Analysis Amplification Kit following the BD Rhapsody System mRNA Whole Transcriptome Analysis (WTA) and Sample Tag Library Preparation Protocol (BD Biosciences). Sequencing was performed in paired-end mode on a NovaSeq 6000 respectively.

### Quality control of the scRNA-seq data and integration of multiple scRNA-seq datasets

Raw sequencing data were de-multiplexed, processed and examined through the BD Rhapsody Whole Transcriptome Analysis pipeline. For

further analysis, the gene expression matrices were analyzed in the R environment. DoubletFinder R software package (version 1.2.2) was used to calculate, infer and delete doublets in each sample, with default parameters. After removing the doublets, the Seurat R software package (version 4.0.0) was used to analyze the filtered gene expression matrix for downstream quality control. Only genes expressed at >3 cells and cells with >200 genes detected were selected for further analyses. Cells that meet the following conditions were defined as low-quality cells and were removed: (i) <500 UMIs (unique molecular identifiers), (ii) <200 or >5000 genes, and (iii) >20% UMIs derived from the mitochondrial genome. After removal of low-quality cells, the R package Seurat (version 4.0.0) was used to integrated multi-sample. Briefly, the function NormalizeData were applied to each expression matrix of AH and TH samples for log-transformation and the function FindVariableFeatures were used to select the top 2000 genes with high cell-to-cell variation. Then, the function FindIntegrationAnchors and IntegrateData were used to identify the "anchors" between individual datasets for obtaining an unbatched dataset of AH and TH group, respectively. Ultimately, we identified 7 major cell types in AH group based on 63,002 cells, and 7 major cell types in TH groups based on 65,850 cells. Then, using the FindIntegrationAnchors and IntegrateData functions to integrate AH and TH group[64], a total of 128,852 cells from 7 major cell types were obtained. Next, we clustered cells using the FindNeighbors and FindClusters functions and performed nonlinear dimensional reduction with the RunUMAP function with default settings.

## Cell type annotation and cluster marker identification

The function "FindAllMarkers" were used to find markers for each of the identified clusters and annotated them according to the expression of canonical markers of particular cell types and the annotation reference created by SingleR (version 1.2.4) based on the published single-cell transcriptome data of immune cells in tissues and peripheral blood.

## Subclustering of major cell types

For each major type, cells were extracted from the overview integrated dataset first. Next, these major cell types were integrated for further subclustering. After integration, genes were scaled to unit variance. Scaling, dimensional reduction, and clustering were performed as described above.

## Expression signature analysis

To evaluate the degree to which individual cells expressed a certain predefined expression gene set, the AddModuleScore function in Seurat with default settings was used to calculate the signature scores. The naïve, cytotoxicity, and exhaustion scores were measured based on well-defined naïve markers (*CCR7*, *TCF7*, *LEF1*, and *SELL*), cytotoxicity-associated genes (*PRF1*, *IFNG*, *GNLY*, *NKG7*, *GZMB*, *GZMA*, *GZMH*, *KLRK1*, *KLRB1*, *KLRD1*, *CTSW*, *CST7*, and *GZMK*), exhausted markers (*LAG3*, *TIGIT*, *PDCD1*, *CTLA4*, and *HAVCR2*), and effector markers (*S100A4*, *ANXA1*, *CD40LG*, *CXCR6*, and *CXCR3*).

## Functional enrichment analysis

Gene set variation analysis (GSVA) was performed using the R package GSVA (version 1.38.2) to estimate the enrichment scores of biological pathways. The "gsva" algorithm gave positive and negative values to positively or negatively regulated pathways respectively. The hallmark and gene ontology (GO) gene sets were obtained using the R package msigdbr (version 7.4.1) and from the MsigDB website (http://software.broadinstitute.org/gsea/msigdb). For the differential functional pathways between naïve CD4 + T cells and regulatory CD4 + T cells of AH and TH through GSVA analysis, we used FindAllMarkers function to identify maker genes for naïve CD4 + T cells and regulatory CD4 + T cells. If the differential pathway contained at least two marker

genes, the pathway would be selected and showed. The differentially expressed genes (DEGs) between the two subtypes of cells were identified using the FindMarkers function of the R package Seurat with the same parameters used for FindAllMarkers. The enrichGO function and enrichKEGG function from the R package clusterProfiler (V.4.4.4), respectively, were used for the analysis of GO terms and KEGG pathways, with pvalueCutoff set to 0.05.

## Pseudotime trajectory analysis

"CellCycleScoring" and "ScaleData" functions in Seurat R software package (version 4.0.0) were used to correct the expression levels of cell cycle of cell subtypes. The R package Monocle2 (version 2.16.0) was sued to infer putative differentiation trajectories of T/NK cell subtypes and B cell subtypes. The gene-cell matrix of UMI counts was provided as the input to Monocle, and then, the newCellDataSet function was employed to create a CellDataSet. The significantly different genes were identified by the differentialGeneTest function.

## Cell-cell communication analysis

The number and strength of intercellular interactions were characterized by the R package CellChat (version 1.1.3) with default parameters. Then, CellphoneDB (version 2.0.0) was used to explore the potential communication between different cell types based on the expression of ligand-receptor pairs. The receptors or ligands expressed by more than 10% of cells in a cluster were included in the downstream analysis. To assess the significance of a ligand-receptor pair between two clusters, an empirical P value was determined by randomly assigning the cluster labels of each cell for 1000 times.

## mNGS and analysis

Extract DNA from each sample using the TruSeq Nano DNA LT Sample Preparation Kit (Illumina), and then cut to approximately 300-500 bp using the Covaris S220 instrument. Construct libraries through end-repair, A-tailing, and adapter ligation. Subsequently, they were sequenced in PE150 mode on the Illumina platform. The metagenomic sequencing and analysis were conducted by OE Biotech Co., Ltd. (Shanghai, China). Reads were trimmed and filtered using Trimmomatic (v0.36). Then, using BBduk and BBmap tools to remove human read contaminations. The software Bowtie2 (v2.2.9) was used for aligning reads, MEGAHIT (v1.1.2) was used for performing metagenome assembly, prodigal (v2.6.3) was used for performing open reading frame prediction in assembled scaffolds, and CDHIT (v4.6.7) was used for building the non-redundant gene sets for all predicted genes. Then translate the nucleotide sequences into amino acid sequences. Bowtie2 (v2.2.9) was used for aligning against the non-redundant gene set (95% identity) and abundance information was determined for genes in each sample. We used the statistical algorithm Bray-Curtis to calculate the distance between pairwise samples, obtained the beta diversity distance matrix, and then perform hierarchical clustering analysis based on the distance matrix, while used the unweighted pair group method with arithmetic mean (UPGMA) algorithm to construct a tree structure. LEfSe was used to identify the microbial taxa that significantly differed between the AH and TH group, which using non-parametric tests to identify significant features, with a threshold LDA score of > 3 used.

## Immunofluorescence and immunohistochemistry

Sections were cut from formalin-fixed/paraffin-embedded samples from the analyzed patients, which were collected from Department of Pathology, Capital Institute of Pediatrics. The antibodies used in the present study were anti-MASA1 (CST, cat#48750t, clone: E7B7T, 1:1000), anti-CD3 (Affiity, cat#DF6594, clone: Polyclonal, 1:4000), anti-IGHG1 (Abcam, cat#ab109489, clone: EPR4421, 1:5000), anti-LYZ (Abcam, cat#ab108508, clone: EPR2994(2), 1:1000), anti-PLD4 (Affiity, cat#DF4294, clone: Polyclonal, 1:2000), anti-KRT18 (Abcam,

cat#ab133263, clone: EPR1626, 1:100), anti-S100A9 (CST, cat#72590t, clone: D5O6O, 1:3000), anti-TIGIT (CST, cat#99567t, clone: E5Y1W, 1:500), anti-PTPRC (CST, cat#13917t, clone: D9M8, 1:20000), anti-CD22 (Abcam, cat#ab207727, clone: EPR20061, 1:10000), anti-XCL1 (Abcam, cat#ab302522, clone: EPR26181-30, 1:100), anti-CCL4 (Abcam, cat#ab45690, clone: EP521Y, 1:500), anti-IGHD (Proteintech, cat#CL488-67538, clone: 1D1B12, 1:300), and anti-CNR2 (SANTA, cat#sc-293188, clone: 3C7, 1:5000). Multiplex immunofluorescence staining was performed using the Opal 7-Color Manual IHC Kit (PerkinElmer, NEL811001KT) according to the manufacturer's protocol. For immunohistochemistry, the sections were incubated with the primary monoclonal antibodies at 4°C overnight. Then, a Power-Vision two-step histological staining reagent (ZSGB-Bio, Anti-rabbit; PV-9001) and 3,3-diamino-benzidine tetrahydrochloride substrate kit (ZSGB-Bio, ZLI-9018) were used to visualize the localization of the antigen according to the manufacturer's instructions.

### OVA-induced allergic rhinitis rat model

Rats of the model group were treated with 0.3 mg OVA as antigen, 30 mg of aluminum hydroxide powder as adjuvant, and 1 mL of normal saline were added to form a suspension. Basic sensitization was performed by intraperitoneal injection once every other day, for a total of 7 times. On the 15th day, rats were taken with their heads down, and 50 μL of 50 mg/mL OVA solution was dropped into each nasal cavity as a stimulus, once daily for 7 consecutive days. From day 22, in order to maintain the pathological symptoms of rats, 50 μL of 10 mg/mL OVA solution was dripped into each nostril daily for 7 consecutive days. Rats of the control group were given physiological saline at the same time. After the completion of all procedures for the model, peripheral blood serum and nasal lavage fluid were collected for cytokine detection using Luminex liquid suspension chip detection (Wayen Biotechnologies, Shanghai, China). Sections were performed on immunofluorescence staining.

### Flow cytometry and ELISA

The single-cell suspension of adenoid tissues and tonsil tissues was obtained through grinding. The experiments were conducted with $2 \times 10^6$ cells per group. After FC receptor blocking, cell viability staining, and membrane permeabilization, the cells were stained with CD45AlexaFluor700 (Biolegend, cat#368514, clone: 2D1, 1:20), CD3BV786 (BD, cat#563800, clone: SK7, 1:20), CD8 FITC (Biolegend, cat#344704, clone: SK1, 1:20), intracellular antibody Granzyme K APC (Biolegend, cat#370510, clone: GM26E7, 1:20), Brilliant Violet 605 anti-human CD197 (CCR7) (Biolegend, cat#353223, clone: G043H7, 1:20), Alexa Fluor 647 anti-human CD11a (Biolegend, cat#301218, clone: HI111, 1:20), PE anti-p53 (Biolegend, cat#645805, clone: DO-7, 1:20), Brilliant Violet 421 anti-human CD284 (TLR4) (Biolegend, cat#312811, clone: HTA125, 1:20), R718 Mouse Anti-Human CD25 (BD, cat#752147, clone: 2A3, 1:20), RB780 Mouse Anti-Human FoxP3 (BD, cat#568682, clone: 259D/C7, 1:20), PE Mouse Anti-FoxP1 (BD, cat#564216, clone: JC12, 1:20), BV421 Mouse Anti-Human RANTES (BD, cat#564754, clone: 2D5, 1:20) and Brilliant Violet 605 anti-human TNF (Biolegend, cat#502935, clone: MAb11, 1:20). The samples were analyzed using the Aurora spectral flow cytometer (Cytek, USA) and data analysis was performed using the SpectroFlo software (Cytek, V3.2.1), with single-stained cells used as reference controls for spectral compensation (Supplementary Fig. 12-14).

For ELISA, tissues were rinsed in ice-cold PBS to remove excess blood thoroughly, minced and weighted to ensure 50 mg of tissue per tube, then homogenized each tube in fresh lysis buffer (Cloud-Clone, IS007-3). After centrifugation, the supernatant was collected and stored at -80°C. Human Granzyme A (GZMA) ELISA kit (Biomatik, cat# EKU04562) was employed to analysis the level of Granzyme A in the supernatants of the tissues.

### Sorting and culture of B cells and T cells

The single-cell suspension of AH and TH was obtained through grinding. the cells were stained with CD45 APC-CY7 (Biolegend, cat#368518, clone: 2D1, 1:20), CD3 APC (Biolegend, cat#300312, clone: HIT3a, 1:20) and CD20 FITC (Biolegend, cat#302304, clone: 2H7, 1:20). B cells and T cells were sorted using BD FACSAria Fusion Flow Cytometer (BD FACSAria SORP). Gating strategy for sorting B cells and T cells in AH and TH. A) Exclude doublets from single cells through the area and height of FSC. B) Using FSC and SSC parameters to remove debris from single cells and selecting complete clusters of target cells. C) Remove dead cells from the target cells by labeling with BD Pharmingen 7-AAD (BD, 559925). D) Select all white blood cells using CD45 antibody in live cells, and E) further select T cells and B cells using T cell specific antibody CD3 and B cell specific antibody CD20, respectively. F) Then, sort these two cell populations. Collect cell supernatant from the cultured B cells and T cells with $ddH_2O$ or LPS (100 ng/ml) for 36 hours (Supplementary Fig. 15). Concentrations of cytokines were measured using Luminex liquid suspension chip detection (Wayen Biotechnologies, Shanghai, China).

### Statistics and reproducibility

GraphPad Prism 10.0 was used for statistical analysis on the Elisa assay results, and all other statistics analyses were performed using the R software (version 4.0.5) and described in the Methods section and Figure legends. Unless otherwise specified, the statistical analyses were performed in a two-sided manner. The statistical significance was assessed using Welch's t-test, Welch's F-test, and Wilcoxon's test. The correlations between variables were computed with Spearman's r and Pearson's r. Data are expressed as the mean ± SD in bar-plots, while are expressed as median, upper and lower quartiles and 1.5× interquartile range in box-plots, and are expressed as maximum, minimum, and density distribution in violin-plots.

For quantification of histology, immunohistochemistry and immunofluorescence. a) For follicular quantification of adenoid tissues at different ages, we randomly selected 6 samples in each age and grade, and randomly selected 3 chosen fields per a sample by microscopy (20X). Measurement of the diameter of follicles was calculated as the average diameter of all follicles in each field by using Aperio ImageScope (version 12.3.3.7014) software. b) For follicular quantification in 12 pairs of samples used for scRNA-seq, we randomly selected 6 chosen fields by microscopy (20X) in each tissue and quantified follicle size using the same method as described above. Grade III of AH, $n = 4$. Grade IV of AH, $n = 8$. Grade II of TH, $n = 11$. Grade III of TH, $n = 1$. For nasopharyngeal lymph nodes in rat model, we measured the diameters of all follicles in the lymph nodes with the largest area ($n = 6$ rats/group/experiment). c) For immunohistochemical quantification of CD20 and CD3, five random fields were selected for each tissue, and the average area of positive area of each tissue was calculated using Analyze-Measure in Image J software ($n = 12$ samples/group). d) For immunofluorescence quantification of MS4A1, CD3E, IGHG1, LYZ, PLD4, KRT18 and S100A9, slides were scanned using PerkinElmer Vectra3 platform and images were quantified using Inform software (V.2.6.0) ($n = 12$ samples/group). e) For co-expression quantification, we used the colocalization finder function in Image J software to calculate the Pearson's correlation of two color fluorescence co-localization. GraphPad Prism 10 was used for all data statistics and result drawing ($n = 12$ samples/group).

### Reporting summary

Further information on research design is available in the Nature Portfolio Reporting Summary linked to this article.

## Data availability

The raw data of scRNA-seq generated in this study have been deposited in the Genome Sequence Archive in National Genomics Data Center database under accession code HRA006738, which can be

publicly accessible at https://ngdc.cncb.ac.cn/gsa-human/browse/HRA006738. The raw sequencing data are available under controlled access due to data privacy laws related to patient consent for data sharing. Access can be obtained by approval via their respective DAC (Data Access Committees) in the GSA-human database, please refer to the detailed guide: https://ngdc.cncb.ac.cn/gsa-human/document/GSA-Human_Request_Guide_for_Users_us.pdf. According to the guidelines of GSA-human, all non-profit researchers are allowed access to the data and the Principle Investigator of any research group is allowed to apply for Controlled-access of the data. For data requests, DAC will respond within two weeks. The data will be available within a week once the access has been granted and they will be available to download for one year. The raw data of mNGS generated in this study have been deposited in the Genome Sequence Archive in National Genomics Data Center database under accession code CRA020419, which can be publicly accessible at https://ngdc.cncb.ac.cn/gsa/browse/CRA020419. All other data are available in the article and its Supplementary files or from the corresponding author upon request. Source data are provided with this paper.

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

## Acknowledgements

We are grateful to Prof. Jun Tai from Department of Otolaryngology, Capital Institute of Pediatrics, for providing AH samples for pre-experiments, and we also thank Dr. Hongping Hou from Institute of Chinese Materia Medica, China Academy of Chinese Medical Sciences for providing rat model samples. We thank for the financial support by grants from the National Natural Science Foundation of China for Key Programs of China Grants (82130065, J.Y.), the Beijing Natural Science Foundation (7222014, J.Y.), Research Foundation of Capital Institute of Pediatrics (CXYJ-2021-01, J.Y.), FENG foundation (FFBR-202103, J.Y.), Beijing High-Level Public Health Technical Talent Project (2023-02-08, J.Y.), Beijing Hospitals Authority's Ascent Plan (DFL20241301, J.Y.), Beijing Municipal Public Welfare Development and Reform Pilot Project for Medical Research Institutes (JYY2023-19, J.Y.) and Beijing Finance Bureau (CIP2024-0040, J.Y.).

## Author contributions

J.Y. and Q.G. designed the study. Q.G. provided guidance on clinical expertise and and cohort data. Z.Y., Z.X., and T.F. performed data analysis and cytological experiments. Z.Y., S.L., J.C. and Y.K. processed the sequencing data, and assisted in the analyses. B.Z., J.L. and C.P. provided clinical samples for the experiment, collected and managed patient prognosis information. R.W., Z.T. and Y.G. performed the sample curation and sample collection. Z.Y., Z.X., B.D., Y.F., H.Z., G.X. and C.Y. prepared the figures and tables. Z.Y., L.G., J.F., Z.F. and S.Y. were involved in proofreading and deep editing and approved the final manuscript. Y.Y., L.H., and S.Z. participated in data interpretation and analysis. J.Y. and Q.G. devised the main conceptual idea and supervised the project, performed proofreading and deep editing of the manuscript, and approved the final draft. All authors contributed to the article and approved the submitted version.

## Competing interests

The authors declare no competing interests.
