## [Transparent Peer Review file · Nature Communications]

Parallel comparison of T cell and B cell subpopulations of adenoid hypertrophy and tonsil hypertrophy of children

Corresponding Author: Dr Jing Yuan

Version 0:

Reviewer comments:

Reviewer #1

(Remarks to the Author)

The authors present their work on the distribution of T and B cell subpopulations in hypertroph adenoids and tonsils. In total, they reported 1.209 individuals between 1 and 15 years of age. Single cell analyses of 12 paired adenoid and tonsil samples revealed various differences, i.e. plasma cells were less frequent and naive B cells were more frequent in "hypertroph adenoids". They also report that the "cytotoxicity" and number of cytotoxic T cells decreased with the extent of hypertrophy in adenoids. They claim that their "atlas" of immune cells contribute to the understanding of the immune system in the respiratory tract. The authors conclude that the different trajectories of tonsils and adenoids throughout life might reflect the divergent composition of B and T cell subpopulations.

In summary, the authors present a thorough characterization of the immune environment in adenoids and tonsils. The work is original and yields novel, though purely descriptive insights into the different biology of hypertroph adenoids and tonsils

comments:

1. The overall study population is very heterogenous and appears to be poorly characterized. At least the patients used for single cell analyses should be characterized more in detail.

Age: The age range (1-15 years) is considerable, reflecting a very immature stage of the lymphatic system (1 year, fewer class switched B cells, lower levels of serum-IgA and IgE) and a more or less adult immune system (15 years, more mature secondary lymphatic tissues). The single cell studies were performed in children between roughly 4 and 6 years.

Infections: Were infections assessed? Were the tissue donors infected or recently infected?

Vaccinations: Were individuals immunized? e.g. the authors state the some infections such as pneumococci elicit characteristic inflammation patterns. Thus, if the individuals were immunized against pneumococci this might also influence the pattern of lymphocyte subpopulations.

Allergies: The potential influence of allergies on hypertroph tonsils/ adenoids might increase with age (virtually absent in 1 year old, starting between 4 and 6 years and probably considerable in 15 year old individuals). Was the medical history known? Doctor's diagnosis of allergies/ allergic rhinoconjunctivitis?

Pollution: Depending on the exposure to indoor and outdoor air pollution, the pathophysiology of hypertroph adenoids of tonsils is likely to differ. Were the individuals exposed to tobacco smoke?

Methods:

2. The degree of hypertrophy was assessed by Brodsky (tonsil) and Parikh (adenoid) grading scales. These scale are of limited reproducibility. Thus to increase reliability, it would have been recommendable to perform the classification by two independent experts and test for reproducibility.

3. The authors perform a multitude of stainings and molecular genetic techniques to characterize the lymphatic tissues of interest. The methods appear to be sound and reproducible and they reflect the current status of knowledge. The methods section gives sufficient information to reproduce the experiments. They accurately mention the limitation that clonal relationships of B and T cell receptor genes were not assessed.

4. The authors should test the hypothesis that the frequencies of B cell and T cell subpopulations or of other variables they tested do not differ between girls and boys.

The manuscript requires thorough language editing.

Reviewer #2

(Remarks to the Author)

In the manuscript, the authors showed the differences in the composition of immune cells between adenoid hypertrophy (AH) and tonsil hypertrophy (TH) using single-cell RNA sequence analysis. These results suggest the differences in host immune responses in human upper respiratory tissue regions.

Waldeyer's lymph ring in rodents does not exactly replace the function of human tonsils. Therefore, the work is vital in showing the immunology of human upper respiratory tissue.

However, I have several concerns about the methodology and data analysis. In addition, the study is mainly based on single-cell analysis, and there is little data based on functional analysis to verify the data on single-cell analysis.

For specific comments, please see below.

Major comments

- 1) The references used as the definition of hypertrophy in adenoid and tonsils are not appropriate. Brodsky L. showed TH classification in *Pediatr Clin North Am.* 1989;36(6):1551-1569. Parikh SR showed AH classification in *Otolaryngol Head Neck Surg.* 2006 Nov;135(5):684-687. Ref 20 is only comments.
- 2) It would be informative to show the number of people, sex ratio, age, and BMI for each of AH+TH+ and AH+TH-. Different grading systems are also used for the tonsils and adenoids, so each stage should be listed.
- 3) Regarding ethical issues, please be assured that no tonsillectomy or adenoidectomy was performed in the TH- or AH- groups. As a related ethical issue, it should be stated whether all hypertrophy diagnoses were pathologically proven (i.e., whether there were any malignant findings).
- 4) If Parikh's grading system takes over the definition of AH, grade 1 is not hypertrophy and should be excluded from AH (or a group of AH-TH+ should be created).
- 5) In Fig 1 f and supply fig 1 g-o, the authors should compare the data separately for AH+TH- and AH+TH+.
- 6) In Fig. 1 and sup Fig. 1, there is no mention of when and how the composition of blood immune cells was collected and analyzed. Is it blood cell measurement, automatic measurement, single cell, or flow cytometry, and if it was taken before surgery, were there any changes after surgery? The authors conclude that the difference in the composition of blood lymphocytes is the result of a local immune response in the upper respiratory tract. Still, it is challenging to conclude without comparing the results of blood tests before and after surgery.
- 7) If the authors would like to compare AH and TH, the tonsils and adenoids should be analyzed separately in Fig. 2, single-cell analysis.
- 8) It is necessary to explain which cells are analyzed in Fig. 2e.
- 9) In Fig 3a, the authors did not show the method used for IHC, the CD20 antibody, or what was used for color development. If it is DAB, the color development time seems long, so please show an appropriate negative control slide to show that the color development time is suitable (one slide is enough).
- 10) Analyzing and showing AH+ and TH+ single-cell data separately in all figures would be informative.
- 11) In the trajectory analysis, please mark the direction with arrows.
- 12) In the discussion part, the tonsils also regress, so the description of the discussion is misleading. The references are not appropriate (32 and 33 are indexed databases that provide medical explanations for adenoids. There is no mention that adenoids regress compared to tonsils).
- 13) Most of the data is based on single-cell analysis, and to verify this, functional analysis of sorted B cells and T cells may be necessary. For example, analysis of antibody induction ability and cytokine production ability against stimulation by foreign antigens, LTA, and LPS.

Reviewer #3

(Remarks to the Author)

In this manuscript, the authors employed single-cell RNA sequencing to investigate the immune microenvironment in children with adenoid hypertrophy (AH) and tonsil hypertrophy (TH), analyzing 12 paired samples. Their findings indicate that AH and TH tissues contain similar proportions of B cells and T/NK cells. However, AH is characterized by a higher percentage of naive B cells, regulatory CD4+ T cells, and effector memory CD4+ T cells with an exhausted phenotype, whereas TH has a larger proportion of plasma B cells.

The manuscript primarily provides a descriptive transcriptome profiling of the immune cell populations in these tissues, speculating on potential functions without including functional experiments to validate the findings.

The figures are sometimes inadequately presented, lacking critical elements such as scales, significance markers, and units. Additionally, many figure legends do not provide sufficient detail for proper interpretation. There are several errors in both the manuscript and the figures (e.g., the title should read "parallel" instead of "parallely"), indicating the need for thorough proofreading.

Key findings are not clearly highlighted, and the final conclusions are weakly presented, often speculative, and not always supported by the data as currently presented. Additional validation experiments, such as comparisons with healthy tissue, would strengthen the manuscript and improve its impact.

While the topic may interest the Nature Communications audience and the patient samples are rare and valuable, substantial revisions are necessary before this work can be considered for publication. Furthermore, the authors have not sufficiently emphasized the novelty of their findings, making it challenging for readers to distinguish between new insights

and existing knowledge. The current understanding of AH and TH is not well-documented, and it is crucial for the authors to underline how their findings differ from or support the existing knowledge.

Major Comments

- Figure 1F: Clarify what the p-value refers to. Is it for the comparison between the last group and the first? This issue also applies to Figure S1F-I. Add "number and units" on the y-axis.
- Figures 1E, 2A, 3A, 4A: What is the purpose of these images? The signal in Figures 3A and 4A and the follicular size in Figure 1E have not been quantified. One or two representative images would suffice (with the others in supplementary information), especially if there is no main message associated with them.
- Figure 2C: The order of column labels is inverted.
- Figure 3B: The shape of the UMAP is the same as for the total cells (Figure 2B). The B cells should be re-plotted as in Figure 4B for T/NK cells.
- Figures 3D-3E: The pseudotime appears biased by the cell cycle, as naive and memory cells (mostly quiescent) are at the beginning of pseudotime, whereas they typically appear later. Mention or correct for these cell cycle biases in the text. This also applies to T cells in Figure 4E (lines 188-191).
- Figure 4B: The colors in the UMAP do not correspond to the legend. The light blue CD4 intermediate cluster is missing, while there are two separate yellow clusters for CD4 inhibitory.
- Comparing Figures 4B and Supplementary Figure 4D: There are doubts about the cell type assignment. Correct the cluster definition as CD8 is expressed in the CD4 naive cluster.
- Lines 195-197: Since the difference is not significant, the authors cannot claim there are more intermediate CD4 T cells.
- Lines 199-217: The GO analysis misses the column bar information and must be normalized to expressed genes to avoid biases in data interpretation. The whole paragraph is mostly speculative without data validation.

Minor Comments

1. The unit is missing from the scale bar in Figures 1F, 2C, 2D, 3C, 4C, 5A, 5C, 5G, 5H, S1A-B-C-D-H, S3B, S4D, and S4E.
2. Line 89: "4-5 years" should be changed to "2-5 years."
3. In the figures, "celltypes" should be corrected to "cell types."
4. Figure 1E: The blue triangle for the years' scale bar is unnecessary, as is the vertical triangle for grade level since it's not a gradient of grades but rather two discrete grades.

7. Figure 2: Remove titles from the figures.
8. Figures 2B, 2C, 2D, and XX: Replace "Plasma" with "Plasma B Cells" as the former is misleading.

10. Figure 2D: The scale bar is missing, and the fluorescence intensity should be quantified.
11. Figure 2E: The color code of patients doesn't seem to match the dots. It might be better to use green dots for AH, yellow for TH, and use the patient's color code for the matching lines. The name of the cells analyzed in each plot should be reported above the plots.

14. Figure 3D: AH and TH should be labeled at the top of the plots.
15. Figure 4: Better define the categories of CD4-intermediate in the text, as they are not a biologically distinct category.

18. Supplementary Figure 4D: Better define NK cells markers.
19. Figure 5A: The percentage cannot exceed 100%.
20. Figures 5G-H: Description of the columns is missing, and the figure legend says "left and right" instead of "top and bottom." Without this information, it is difficult to understand the results.
21. Figures 4G-H and 5G-H: Significant changes in p-values should be reported to support the conclusions of the data analysis.
22. Figure 6A: This analysis would be more meaningful on differentially expressed genes between AH and TH. The same applies to Figures B and C.
23. Figure 6D: Data must be normalized as percentages since different patients have different numbers of sequenced cells.
24. Figures 6F-H: Quantify the immunofluorescent images to draw a conclusion.
25. Figure S2E: The scale bar is between -1 and 0. Is this correct? What is the unit of average expression?
26. Figures S3B/S4A: There are too many patients for one plot. Either make one plot per patient or remove this plot (as the information is somewhat included in Figure S2A).
27. Line 103: "children s" should be corrected.
28. Line 104: Explain what "the ratio" means in the text. Additionally, there is no panel G in Figure 1.
29. Lines 105-106: Clarify the concept of "being important to infection through immunological responses." The same applies to lines 132-133.
30. Lines 108-109: Rephrase these lines. The aim of the project is to describe the transcriptional status during disease.
31. Lines 124-125: Rephrase these lines. The percentages likely refer to "the total number of cells in each tissue" rather than "the total number of B cells."
32. Line 169: Highlight the significance of the difference between AH and TH T/NK cells in the figure.
33. Line 175: Better discuss the distinction between intermediate and proliferating CD4 cells.

35. Line 193: Modify "The that of regulatory..." to "the percentage of..."

37. Line 240: "Figure 5G" should read "Figure 5g" for consistency.
38. Lines 71-72: These conclusions seem to contrast with lines 311-313. Rephrase these lines for clarity.

Reviewer #4

(Remarks to the Author)

Version 1:

Reviewer comments:

Reviewer #1

(Remarks to the Author)

The authors have successfully addressed all reviewer's comments.

Reviewer #2

(Remarks to the Author)

Following peer review, the authors revised the findings and added functional analysis.

Additional functional analysis includes rat data. However, some of the answers are scientifically questionable and the methods used by the authors might not be appropriate. In particular, it has been anatomically and immunologically pointed out that rats and other rodents do not have tissue equivalent to the human tonsils, so the use of rats is not appropriate. Additionally, batch consolidation in single-cell analysis may be inappropriate.

My several concerns about the methodology and data analysis are below.

1) The references used as the definition of hypertrophy in adenoid and tonsils are not appropriate.

Response:

Thanks for the reviewer's constructive suggestion. We have revised the references on the grade of AH and TH (please see Pages 4-5, lines 79-81).

Comments: In the authors' revised version, the TH grade has been removed from the table, which is no longer informative. The grade of TH should be added to the vertical axis of Table 1 to show how much of each grade of TH is present in each AH and whether there is a correlation between the grade of AH and TH.

2) It would be informative to show the number of people, sex ratio, age, and BMI for each of AH+TH+ and AH+TH-. Different grading systems are also used for the tonsils and adenoids, so each stage should be listed.

Response:

Thanks for the reviewer's constructive comments. We redefined AH grade I as AH-, and showed the number, sex ratio, age, and BMI of patients with AH grade I-IV in Table 1. The number, sex ratio, age, and BMI of patients in AH-TH-, AH+TH-, AH-TH+, AH+TH+ groups are shown in Table 2.

Comments: Table 2 shows that the range of BMI for each group was very wide. Therefore, it is necessary to show the percentage of underweight or overweight children based on BMI. The normal range for BMI in children varies with age and sex. The definition of overweight or underweight showed at CDC homepage (<https://www.cdc.gov/bmi/child-teen-calculator/bmi-categories.html>).

In addition to that the accompanying diseases of OSAS shown in lines 53-54 of the introduction are reports of adults, and OSAS in children has been reported to be accompanied by growth disorders (low BMI) and restlessness etc., which is consistent with the information provided by the authors especially in the AH+TH+ group. It would be informative that lines 53-54 of the introduction need to be revised according to the information in table 2 (The references should be changed to include comorbidities in pediatric OSAS).

6) In Fig. 1 and sup Fig. 1, there is no mention of when and how the composition of blood immune cells was collected and analyzed. Is it blood cell measurement, automatic measurement, single cell, or flow cytometry, and if it was taken before surgery, were there any changes after surgery? The authors conclude that the difference in the composition of blood lymphocytes is the result of a local immune response in the upper respiratory tract. Still, it is challenging to conclude without comparing the results of blood tests before and after surgery.

Response:

The peripheral blood indexes were obtained by performing complete blood counts on the patients during the consultation, which were obtained through Flow Cytometry method with semiconductor laser on Automated hematology analyzer Sysmex (please see Pages 19-20, lines 410-417). We counted the indexes of immune cells. At present, postoperative recovery is not based on changes in peripheral blood indexes, so patients would not receive complete blood counts during follow-up visits, which were in line with diagnostic and treatment standards. Therefore, we have added the AH-TH- group as a healthy control and revised the conclusion of this section (please see Pages 5-6, lines 95-103, Table 2 and Supplementary Fig. 1d-

g).

Comments: In the end, there is no clear statement as to when the blood was taken except for grade I patients who did not undergo surgery, whether before or after the surgery, and I believe that my questions are not answered yet.

7) If the authors would like to compare AH and TH, the tonsils and adenoids should be analyzed separately in Fig. 2, single-cell analysis.

Response:

Adenoids and tonsils are both gland tissues that play an immune function, and their cell composition is similar. The integration analysis of 24 sample data from 12 patients could help remove the batch effect, better normalize the expression level of genes, and help to identify the cell types and gene expression. At present, the R software package Seurat also recommends the integration analysis of multiple tissues and samples, and has launched a variety of methods to integrate gene expression. Our data use the FindIntegration Anchors function and IntegrateData function recommended by the software package to integrate the sample data, which could better present the cell type and gene expression of each sample. Moreover, Supplementary Fig. 2h shows the UMAP map of cell types of AH and TH, respectively. Supplementary Fig. 2j shows the marker gene expression of cell types from AH and TH, respectively.

10) Analyzing and showing AH+ and TH+ single-cell data separately in all figures would be informative.

Response:

Thanks for the reviewer's constructive suggestion. The UMAP maps of the cellular distribution of AH and TH were shown in Supplementary Fig. 2h, Supplementary Fig. 4b and Supplementary Fig. 6b. Moreover, we have added the marker gene expression of main cell types, B cell subtypes and T cell subtypes of AH and TH (please see Supplementary Fig. 2j, Supplementary Fig. 4e and Supplementary Fig. 6f) in the revised manuscript.

Comments: The methods section states that the FindIntegration Anchors function and IntegrateData function were used for individual datasets (not between tissues).

Furthermore, based on the method, it can be interpreted that the authors performed single-cell analysis by grinding adenoid and tonsil tissues together from the beginning (before applying the samples BD Rhapsody). Batch consolidation is typically performed based on data from independent applying, especially for tissues such as adenoids where single cell analysis cell characterization of tissues has not yet been established (bioRxiv, PMID: 39484391).

Again, please state in the method whether the adenoids and tonsils were mixed to create a cell suspension before application. If the analysis uses a premixed cell suspension, the methodology is scientifically questionable. If not, analyze the samples separately and then combine the data to compare. Analysis and consideration is required before and after the integration.

13) Most of the data is based on single-cell analysis, and to verify this, functional analysis of sorted B cells and T cells may be necessary. For example, analysis of antibody induction ability and cytokine production ability against stimulation by foreign antigens, LTA, and LPS.

Response:

Thanks for the reviewer's constructive suggestion. We have added functional analysis of B and T cells in AH and TH, and collected 6 pairs of additional AH and TH samples, including 3 cases of AH grade III, 3 cases of AH grade IV, and 9 cases of TH grade II (Supplementary Table 3). And we have also collected peripheral blood from children who came to the hospital for physical examination as a healthy control. Using antibodies CD20 and CD3 respectively, B cells and T cells were sorted from tissues and peripheral blood by flow cytometry. The results showed that compared to TH, AH contained a higher proportion of B cells and fewer T cells (Fig. 5e). After 36 hours of in vitro culture, cytokine detection was performed on the cell supernatant. Through evaluating the levels of cytokines that perform different functions, we determined the main functions of B and T cells in AH and TH in the revised manuscript. Specifically, compared to TH, T cells in AH had more significant immune suppression, chemotaxis, and apoptosis, while B cells had weaker antigen presentation and proliferation abilities. Notably, the stimulation of LPS amplified the difference in secretion function between two cell types in two tissues (please see Pages 13-14, lines 262-284).

Comments: Please show the gating strategy for each cell sorting. Without that gating data, it is impossible to determine whether the method is scientifically valid.

It has been suggested that immune cells in AH are more exhausted than those in TH, but there is insufficient discussion of this finding (Differences in the functions, cellular composition, characteristic feature and epithelial characterization of the two tissues etc.).

Major Comments

Comments: The authors added rat experiments based on a report by Anqi Liu et al. Unfortunately, their report examined hypertrophy of the NALT. Tonsillar tissue does not exist in rats, mice, or hamsters. It remains to be seen whether rodent NALT is an anatomically and immunologically identical tissue to the human tonsil. This model does not reflect adenoid or tonsillar hypertrophy (PMID: 21869895 Casteleyn C, et al., 2011, Clin Dev Immunol).

Comments: Throughout the manuscript, the terms adenoid hypertrophy and adenoid (or tonsil hypertrophy and tonsil) are used interchangeably, which can be confusing. Specifically, AH and TH in Line 429 and Line 430 should be described as adenoid (tissue) and tonsil (tissue), respectively. It is important to note that AH and TH are conditions and not tissue or the entities themselves. The same goes for Line 523, 539.

Reviewer #3

(Remarks to the Author)

While this reviewer appreciates the improvements made to the manuscript and figures, the response to some of my initial comments remains inadequate and misleading.

For instance, (lines 258-284) the authors have included peripheral blood (PB) from healthy controls to strengthen their conclusions, but they have compared PB cells from healthy controls to AH and TH cells from patients. What is presented as of now, is not a straightforward comparison between healthy and diseased conditions, but rather a comparison between different tissue sources. It is well established that immune cells vary significantly depending on the tissue in which they reside.

Moreover, the authors did not normalize the data. At the very least, they should have set the control values in Figure 5F to "1" to emphasize the differences between pre- and post-LPS treatment. The authors state that "after LPS treatment... cytokine secretion of B cells and T cells in AH and TH was more vigorous." However, normalization reveals a different picture. For instance, IL-8 increases approximately fivefold in controls (from ~1 to ~5), whereas the increase in AH and TH is only between 1.2 and 2 times. This reviewer would conclude the opposite of the authors' statement. The same applies to G-CSF, MIP-1a, MIP-1b and maybe to others.

Under this reviewer's suggestion, the authors corrected the pseudotime of B cells, since memory B cells were strangely at the beginning of pseudotime. However, it is not so clear how they perform this normalization, they have just shown the percentages of cells in the different cell cycle stages. The authors affirmed they "correct by changing the gene expression value and by <<adding>> some gene expression". What does it mean to "add" some gene expression? Overall, this reviewer does not think they solved this issue. The authors have just renamed the clusters of memory B cells under the name of "atypical memory B cells", but they did not detect clusters of memory B cells in the plot (Figure 3E). To our knowledge atypical memory cells should co-exist with memory cells, not substitute for them. The general pattern of differentiation of B cells is not clear from their data. The authors should try to explain this phenomenon or remove this analysis.

Moreover in Fig4E all CD4/CD8/NK cells have been analyzed in the same plot. There is no meaning in putting different cells from different differentiation processed in the same pseudotime. As CD4 won't convert into a CD8 cell (lines 217-220 must be corrected).

Most of the enriched GSVA terms in CD4-Naive and Treg (Figure 5A-B) are generic and do not underline significant functions/roles of these cells between AH and TH tissues.

The conclusions drawn from that plot are lengthy somewhat meaningless.

Minor comments

Fig 4D panel of TNF, the scale bar is different in the upper facs plot with respect to the bottom. Moreover the facs plot suggests a bimodal distribution, not quantified in the MFI.

"Lefse" at lane 125 must be explained

The metagenomics analysis is missing from the Material and Methods Section.

Fig s2e the enrichment seems to be opposite with the data presented in Fgi S2c/d. For instance Bacteroidetes is enriched in AH in Fig S2C, but the enrichment is in TH in Fig S2e (and so on).

Lines 230-235 can be summarized for clarity

Reviewer #4

(Remarks to the Author)

Version 2:

Reviewer comments:

Reviewer #2

(Remarks to the Author)

Although this reviewer appreciates the improvements made to the manuscript and figures, this reviewer believes that the overall conception of the paper is unclear.

Specifically, is the theme of this study about the "hypertrophy pathogenesis" or "the differences in immune responses between adenoids and tonsils"?

According to the paper PMID: 38301653 (cited by the authors) and the preprint paper PMID: 39484391, the cellular composition of the adenoid and tonsil is markedly different. In particular, the characteristics of epithelial cells are notably distinct. However, this study barely mentions the cluster of epithelial cells.

There have been some concerns about the methodology of single-cell analysis in the past, and this reviewer has confirmed the method during previous peer reviews. Upon further review, this reviewer found that the authors' study did not demonstrate entirely significant differences between the cell clusters of the adenoid (AH) and the tonsil (TH).

Since epithelial cells are considered extremely important as triggers of immune responses, it seems complicated to overlook this cluster. It is necessary to discuss the differences from previous reports thoroughly.

The strength of single-cell analysis is that it is possible to analyze rare cell groups, so the small number of cells is not a limitation.

Reviewer #3

(Remarks to the Author)

The authors have addressed all my comments

Reviewer #4

(Remarks to the Author)

Comments from Reviewer 1

Remarks to the Author:

The authors present their work on the distribution of T and B cell subpopulations in hypertroph adenoids and tonsils. In total, they reported 1.209 individuals between 1 and 15 years of age. Single cell analyses of 12 paired adenoid and tonsil samples revealed various differences, i.e. plasma cells were less frequent and naive B cells were more frequent in "hypertroph adenoids". They also report that the "cytotoxicity" and number of cytotoxic T cells decreased with the extent of hypertrophy in adenoids. They claim that their "atlas" of immune cells contribute to the understanding of the immune system in the respiratory tract. The authors conclude that the different trajectories of tonsils and adenoids throughout life might reflect the divergent composition of B and T cell subpopulations.

In summary, the authors present a thorough characterization of the immune environment in adenoids and tonsils. The work is original and yields novel, though purely descriptive insights into the different biology of hypertroph adenoids and tonsils

1. The overall study population is very heterogenous and appears to be poorly characterized. At least the patients used for single cell analyses should be characterized more in detail.

Age: The age range (1-15 years) is considerable, reflecting a very immature stage of the lymphatic system (1 year, fewer class switched B cells, lower levels of serum-IgA and IgE) and a more or less adult immune system (15 years, more mature secondary lymphatic tissues). The single cell studies were performed in children between roughly 4 and 6 years.

Infections: Were infections assessed? Were the tissue donors infected or recently infected?

Vaccinations: Were individuals immunized? e.g. the authors state the some infections such as pneumococci elicit characteristic inflammation patterns. Thus, if the individuals were immunized against pneumococci this might also influence the pattern of lymphocyte subpopulations.

Allergies: The potential influence of allergies on hypertroph tonsils/ adenoids might increase with age (virtually absent in 1 year old, starting between 4 and 6 years and probably considerable in 15 year old individuals). Was the medical history known? Doctor's diagnosis of allergies/ allergic rhinoconjunctivitis?

Pollution: Depending on the exposure to indoor and outdoor air pollution, the pathophysiology

of hypertroph adenoids of tonsils is likely to differ. Were the individuals exposed to tobacco smoke?

Response:

We thank the reviewer for his/her constructive comments on our observations in the present study. We further analyzed the characteristics of the patients used for single cell analyses in detail. We here address the reviewer's particular concerns point by point as follows.

1) **Age:** The AH patients' ages were concentrated in 2-5 years (please see Page 5, lines 83-84). Therefore, we selected patients aged 2-5 years for single-cell RNA sequencing to identify transcriptional differences between AH and TH (please see Supplementary Table 1).

2) **Infection:** Metagenomic sequencing was then performed on 12 AH and TH tissues to evaluate their tissue infection. We found that the microbial compositions of AH and TH were roughly the same, but AH contained a higher proportion of Bacteroidetes and Deferribacters than that of TH (Supplementary Fig. 2c). At the species level, AH contained a higher proportion of Desulfovibrio and Lachnospiraceae, while TH contained more Haemophilus influenzae (Supplementary Fig. 2d). Moreover, AH contained more bacilli, while TH contained more cocci through LEfSe difference analysis (Supplementary Fig. 2e) (please see Pages 6-7, lines 119-126).

3) **Vaccinations:** With the popularization of vaccination publicity, children have been immunized against pneumococci before the age of 2 years (please see Supplementary Table 1).

4) **Allergies:** 12 patients were examined for allergies before surgery. P01 and P02 had significant allergies to Wormwood, P07 had significant allergies to dog hair, and P08 was allergic to Mould. No obvious allergies had been found in other patients. Meanwhile, allergies did not affect the proportion of main cell types, B cell subtypes and T cell subtypes of AH and TH (please see Supplementary Fig. 3f,g, Supplementary Fig. 5c,d, Supplementary Fig. 7e,f).

5) **Pollution:** At present, our city has strict smoking bans and the quality of parents has been improved, so children have not been exposed to smoke (please see Supplementary Table 1).

Methods:

2. The degree of hypertrophy was assessed by Brodsky (tonsil) and Parikh (adenoid) grading scales. These scale are of limited reproducibility. Thus to increase reliability, it would have

been recommendable to perform the classification by two independent experts and test for reproducibility.

Response:

Thanks for the reviewer's reminder. All patients' AH grades were determined based on their nasal endoscopy results. Specifically, if the area of adenoid obstruction in the nostrils was less than 25%, it was classified as Grade I; if the obstruction area was between 26% and 50%, it was classified as Grade II; if the obstruction area was between 51% and 75%, it was classified as Grade III; and if the obstruction area was between 76% and 100%, it was classified as Grade IV. The specific situation was shown in the following figure. The grade of TH was determined by examination of the throat with a tongue depressor by two ENT clinicians.

Fig. Representative images of nasal endoscopes with different AH grade.

3. The authors perform a multitude of stainings and molecular genetic techniques to characterize the lymphatic tissues of interest. The methods appear to be sound and reproducible and they reflect the current status of knowledge. The methods section gives sufficient information to reproduce the experiments. They accurately mention the limitation that clonal relationships of B and T cell receptor genes were not assessed.

Response:

Thanks for the reviewer's overall evaluation of our manuscript.

4. The authors should test the hypothesis that the frequencies of B cell and T cell subpopulations or of other variables they tested do not differ between girls and boys.

Response:

Many thanks for the suggestion. We found that gender did not affect the proportion of main cell types, B cell subtypes and T cell subtypes of AH and TH (please see Supplementary Fig. 3d, e, Supplementary Fig. 5a, b, Supplementary Fig. 7c, d). Notably, the gender used for statistics refers to the biological sex of a baby at birth.

The manuscript requires thorough language editing.

Response:

We have thoroughly language edited our manuscript.

Reviewer #2 (Remarks to the Author)

In the manuscript, the authors showed the differences in the composition of immune cells between adenoid hypertrophy (AH) and tonsil hypertrophy (TH) using single-cell RNA sequence analysis. These results suggest the differences in host immune responses in human upper respiratory tissue regions.

Waldeyer's lymph ring in rodents does not exactly replace the function of human tonsils. Therefore, the work is vital in showing the immunology of human upper respiratory tissue. However, I have several concerns about the methodology and data analysis. In addition, the study is mainly based on single-cell analysis, and there is little data based on functional analysis to verify the data on single-cell analysis.

For specific comments, please see below.

Major comments

1) The references used as the definition of hypertrophy in adenoid and tonsils are not appropriate. Brodsky L. showed TH classification in *Pediatr Clin North Am.* 1989;36(6):1551-1569. Parikh SR showed AH classification in *Otolaryngol Head Neck Surg.* 2006 Nov;135(5):684-687. Ref 20 is only comments.

Response:

Thanks for the reviewer's constructive suggestion. We have revised the references on the grade of AH and TH (please see Pages 4-5, lines 79-81).

2) It would be informative to show the number of people, sex ratio, age, and BMI for each of AH+TH+ and AH+TH-. Different grading systems are also used for the tonsils and adenoids, so each stage should be listed.

Response:

Thanks for the reviewer's constructive comments. We redefined AH grade I as AH⁻, and showed the number, sex ratio, age, and BMI of patients with AH grade I-IV in Table1. The number, sex ratio, age, and BMI of patients in AH⁻TH⁻, AH⁺TH⁻, AH⁻TH⁺, AH⁺TH⁺ groups are shown in Table2.

3) Regarding ethical issues, please be assured that no tonsillectomy or adenoidectomy was performed in the TH- or AH- groups. As a related ethical issue, it should be stated whether all hypertrophy diagnoses were pathologically proven (i.e., whether there were any malignant findings).

Response:

The AH⁻ and TH⁻ group patients were both from outpatient clinics and have not undergone surgery. The AH and TH tissues of patients undergoing adenoidectomy with tonsillectomy were judged by two or more clinical pathologists, and malignant changes were ruled out. We have checked the pathology diagnosis reports of patients diagnosed with AH and/or TH, and none of them indicated the presence of malignant lesions in the tissues.

4) If Parikh's grading system takes over the definition of AH, grade 1 is not hypertrophy and should be excluded from AH (or a group of AH-TH+ should be created).

Response:

Thanks for the reviewer's constructive suggestion. We have redefined AH grade I as AH⁻, and re-

described the immune cells in the peripheral blood of AH⁻TH⁻, AH⁺TH⁻, AH⁻TH⁺, AH⁺TH⁺ groups of patients in the revised manuscript (please see Pages 5-6, lines 95-103, Table 2 and Supplementary Fig. 1d-g).

5) In Fig1 f and supply fig1 g-o, the authors should compare the data separately for AH⁺TH⁻ and AH⁺TH⁺.

Response:

Thanks for the reviewer's constructive suggestion. By depicting the peripheral blood indexes of patients with AH grade I-IV with/without TH, we found that in AH grade III-IV, compared with TH⁻ patients, there were lower blood lymphocyte number and lymphocyte ratio, but higher neutrophil number and neutrophil ratio in TH⁺ patients (please see Page 6, lines 103-109 and Supplementary Fig. 1h-j).

6) In Fig. 1 and sup Fig. 1, there is no mention of when and how the composition of blood immune cells was collected and analyzed. Is it blood cell measurement, automatic measurement, single cell, or flow cytometry, and if it was taken before surgery, were there any changes after surgery? The authors conclude that the difference in the composition of blood lymphocytes is the result of a local immune response in the upper respiratory tract. Still, it is challenging to conclude without comparing the results of blood tests before and after surgery.

Response:

The peripheral blood indexes were obtained by performing complete blood counts on the patients during the consultation, which were obtained through Flow Cytometry method with semiconductor laser on Automated hematology analyzer Sysmex (please see Pages 19-20, lines 410-417). We counted the indexes of immune cells. At present, postoperative recovery is not based on changes in peripheral blood indexes, so patients would not receive complete blood counts during follow-up visits, which were in line with diagnostic and treatment standards. Therefore, we have added the AH⁻TH⁻ group as a healthy control and revised the conclusion of this section (please see Pages 5-6, lines 95-103, Table 2 and Supplementary Fig. 1d-g).

7) If the authors would like to compare AH and TH, the tonsils and adenoids should be analyzed separately in Fig. 2, single-cell analysis.

Response:

Adenoids and tonsils are both gland tissues that play an immune function, and their cell composition is similar. The integration analysis of 24 sample data from 12 patients could help remove the batch effect, better normalize the expression level of genes, and help to identify the cell types and gene expression. At present, the R software package Seurat also recommends the integration analysis of multiple tissues and samples, and has launched a variety of methods to integrate gene expression. Our data use the FindIntegration Anchors function and IntegrateData function recommended by the software package to integrate the sample data, which could better present the cell type and gene expression of each sample¹. Moreover, Supplementary Fig. 2h shows the UMAP map of cell types of AH and TH, respectively. Supplementary Fig. 2j shows the marker gene expression of cell types from AH and TH, respectively.

8) It is necessary to explain which cells are analyzed in Fig. 2e.

Response:

Sorry for our negligence, we have added the name of cell types in Fig. 2e.

9) In Fig3a, the authors did not show the method used for IHC, the CD20 antibody, or what was used for color development. If it is DAB, the color development time seems long, so please show an appropriate negative control slide to show that the color development time is suitable (one slide is enough).

Response:

All immunohistochemistry sections borrowed from the Department of Pathology, Capital Institute of Pediatrics, Beijing, China, which were used for clinical pathological diagnosis. We believed that the status of these sections was not due to excessive DAB-stained time, but rather due to high

expression levels of CD20. Because there was no nonspecific staining of CD20 negative regions. Furthermore, we added negative controls for DAB-stained of AH, as showed in the figure below:

Fig. Representative images of DAB-stained negative control of AH tissue. Scale bar: 200µm.

10)Analyzing and showing AH+ and TH+ single-cell data separately in all figures would be informative.

Response:

Thanks for the reviewer’s constructive suggestion. The UMAP maps of the cellular distribution of AH and TH were shown in Supplementary Fig. 2h, Supplementary Fig. 4b and Supplementary Fig. 6b. Moreover, we have added the marker gene expression of main cell types, B cell subtypes and T cell subtypes of AH and TH (please see Supplementary Fig. 2j, Supplementary Fig. 4e and Supplementary Fig. 6f) in the revised manuscript.

11) In the trajectory analysis, please mark the direction with arrows.

Response:

We have added arrows in Fig.3e, Fig.4e, Supplementary Fig. 4g and Supplementary Fig. 6h in the revised manuscript.

12) In the discussion part, the tonsils also regress, so the description of the discussion is misleading. The references are not appropriate (32 and 33 are indexed databases that provide medical explanations for adenoids. There is no mention that adenoids regress compared to

tonsils).

Response:

Thanks for the reviewer's constructive suggestion. We have revised this section and its references (please see Page 17, lines 353) in the revised manuscript.

13) Most of the data is based on single-cell analysis, and to verify this, functional analysis of sorted B cells and T cells may be necessary. For example, analysis of antibody induction ability and cytokine production ability against stimulation by foreign antigens, LTA, and LPS.

Response:

Thanks for the reviewer's constructive suggestion. We have added functional analysis of B and T cells in AH and TH, and collected 6 pairs of additional AH and TH samples, including 3 cases of AH grade III, 3 cases of AH grade IV, and 9 cases of TH grade II (Supplementary Table 3). And we have also collected peripheral blood from children who came to the hospital for physical examination as a healthy control. Using antibodies CD20 and CD3 respectively, B cells and T cells were sorted from tissues and peripheral blood by flow cytometry. The results showed that compared to TH, AH contained a higher proportion of B cells and fewer T cells (Fig. 5e). After 36 hours of *in vitro* culture, cytokine detection was performed on the cell supernatant. Through evaluating the levels of cytokines that perform different functions, we determined the main functions of B and T cells in AH and TH in the revised manuscript. Specifically, compared to TH, T cells in AH had more significant immune suppression, chemotaxis, and apoptosis, while B cells had weaker antigen presentation and proliferation abilities. Notably, the stimulation of LPS amplified the difference in secretion function between two cell types in two tissues (please see Pages 13-14, lines 262-284).

References:

1. Hao Y, *et al.* Integrated analysis of multimodal single-cell data. *Cell* **184**, 3573-3587.e3529 (2021).

Reviewer #3 (Remarks to the Author)

In this manuscript, the authors employed single-cell RNA sequencing to investigate the immune microenvironment in children with adenoid hypertrophy (AH) and tonsil hypertrophy (TH), analyzing 12 paired samples. Their findings indicate that AH and TH tissues contain similar proportions of B cells and T/NK cells. However, AH is characterized by a higher percentage of naive B cells, regulatory CD4⁺ T cells, and effector memory CD4⁺ T cells with an exhausted phenotype, whereas TH has a larger proportion of plasma B cells.

The manuscript primarily provides a descriptive transcriptome profiling of the immune cell populations in these tissues, speculating on potential functions without including functional experiments to validate the findings.

Response:

Thanks for the reviewer's constructive suggestion. We have enhanced the credibility of our manuscript through the following experiments:

i) To evaluate the differences in immune efficacy between T cells and B cells of AH and TH, we collected 6 pairs of additional AH and TH samples, including 3 cases of AH grade III, 3 cases of AH grade IV, and 9 cases of TH grade II (Supplementary Table 3). And we have also collected peripheral blood from children who came to the hospital for physical examination as a healthy control. Using antibodies CD20 and CD3 respectively, B cells and T cells were sorted from tissues and peripheral blood by flow cytometry. The results showed that compared to TH, AH contained a higher proportion of B cells and fewer T cells (Fig. 5e). After 36 hours of *in vitro* culture, cytokine detection was performed on the cell supernatant. Through evaluating the levels of cytokines that perform different functions, we determined the main functions of B and T cells in AH and TH. Specifically, compared to TH, T cells in AH had more significant immune suppression, chemotaxis, and apoptosis, while B cells had weaker antigen presentation and proliferation abilities. Notably, the stimulation of LPS amplified the difference in secretion function between two cell types in two tissues (please see Pages 13-14, lines 262-284).

ii) To further demonstrate the proportion and functional differences of naïve CD4⁺ T cells and

regulatory CD4⁺ T cells between AH and TH, we collected 9 pairs of additional AH and TH samples, including 4 cases of AH grade III, 5 cases of AH grade IV, and 9 cases of TH grade II (Supplementary Table 2). The naïve CD4⁺ T cells were selected by flow cytometry using antibodies CD45, CD3, CD4, and CD45RA, while regulatory CD4⁺ T cells were selected by flow cytometry using antibodies CD45, CD3, CD4, and CD25. The results showed that the proportion of naïve CD4⁺ T cells in AH was lower than that in TH, while the proportion of regulatory CD4⁺ T cells in AH was higher than that in TH (Fig. 4g,h). We further examined the expression of core genes of differential expression pathways obtained through GSEA analysis in naïve CD4⁺ T cells and regulatory CD4⁺ T cells of AH and TH using flow cytometry. Based on the differential pathways and the expression of core genes, we speculated that compared to TH, naïve CD4⁺ T cells in AH had weaker proliferation and activation abilities, while regulatory CD4⁺ T cells had stronger chemotaxis and immunosuppressive abilities (please see Pages 12-13, lines 238-257).

The figures are sometimes inadequately presented, lacking critical elements such as scales, significance markers, and units. Additionally, many figure legends do not provide sufficient detail for proper interpretation. There are several errors in both the manuscript and the figures (e.g., the title should read "parallel" instead of "parallely"), indicating the need for thorough proofreading.

Response:

Sorry for our negligence, we have added scale bars and units in the figures, increased the readability of the figure legends, quantified the results of H&E-, immunohistochemistry-, and immunofluorescence-stained sections. And the language of the manuscript has been thoroughly proofread in the revised manuscript.

Key findings are not clearly highlighted, and the final conclusions are weakly presented, often speculative, and not always supported by the data as currently presented. Additional validation experiments, such as comparisons with healthy tissue, would strengthen the manuscript and improve its impact.

Response:

Thanks for the reviewer's constructive suggestion. Because patients without AH would not undergo adenoidectomy surgery, we could not obtain healthy tissue as a control. Therefore, in order to improve the quality of our manuscript, we conducted the following experiments:

i) It had been reported that the nasopharyngeal lymph nodes of OVA-induced adenotonsillar hypertrophy rat model could reflect the pathological changes of human adenotonsillar hypertrophy¹. Therefore, we constructed this rat model and supplemented the methods in Page 25, lines 533-544. Through the evaluation of inflammatory factors in peripheral serum and nasal lavage fluid of rats, we found that there was a significant inflammatory response in the body of the model rats. Compared to the control group, the follicles in the nasopharyngeal lymph nodes of model group were enlarged, while the numbers of B cells, T&NK cells, myeloid cells, plasma cells like dendritic cells, and neutrophils increased. These results indicated that the occurrence of AH and TH increases the infiltration of immune cells in tissues (please see Page 8, lines 148-158).

ii) We collected peripheral blood from children who came to the hospital for physical examination as a healthy control to assess cytokine secretion in B and T cells. After 36 hours of in vitro culture, cytokine detection was performed on the cell supernatant. The results showed that the levels of G-CSF, IL-6, IL-8, MIP-1a, MIP-1b, IL-16, MIF, and SCGF-b in T and B cells of AH and TH were higher than those in the healthy control group. Furthermore, after LPS treatment, the expression of G-CSF, IL-6, IL-8, MIP-1 α , and MIP-1 β was upregulated in B and T cells, while the expression of IL-16, MIF, and SCGF- β was downregulated. These suggest that compared to healthy controls, the cytokine secretion of B cells and T cells in AH and TH was more vigorous, participating in more immune regulation (please see Pages 13-14, lines 262-284).

While the topic may interest the Nature Communications audience and the patient samples are rare and valuable, substantial revisions are necessary before this work can be considered for publication. Furthermore, the authors have not sufficiently emphasized the novelty of their findings, making it challenging for readers to distinguish between new insights and existing knowledge. The current understanding of AH and TH is not well-documented, and it is crucial for the authors to underline how their findings differ from or support the existing knowledge.

Response:

Thanks for the reviewer's reminder. We have added a description of the limitations of single-cell transcriptome studies in AH and TH in the **Introduction** section (please see Page 4, lines 61-63). Moreover, in the **Discussion** section, a summary explanation of our research results has been provided (please see Page 17, lines 353-363) in the revised manuscript.

Major Comments

- Figure 1F: Clarify what the p-value refers to. Is it for the comparison between the last group and the first? This issue also applies to Figure S1F-I. Add "number and units" on the y-axis.

Response:

The *P*-value was obtained through Welch's *F*-test, and $P < 0.05$ indicates that the means of the four groups are not equal. The means of the four groups have been marked in the graph, and the results could show the increasing or decreasing relationship between blood cell results and AH grading. In addition, we have supplemented the *P*-values of pairwise comparisons of AH grading using Welch's *t*-test, and added units for immune cell indexes (please see Fig. 1f and Supplementary Fig. 1c) in the revised manuscript.

- Figures 1E, 2A, 3A, 4A: What is the purpose of these images? The signal in Figures 3A and 4A and the follicular size in Figure 1E have not been quantified. One or two representative images would suffice (with the others in supplementary information), especially if there is no main message associated with them.

Response:

Thanks for the reviewer's reminder. We have conducted quantitative analysis on the results of Fig. 1e, 2a, 3a, and 4a in the revised manuscript.

i) Fig. 1e showed pathological sections of AH patients aged 1 to 15 years old. Compared to AH grade III, the follicle size of AH grade IV were larger and gradually decreased after reaching the maximum at the age of 7 (please see Supplementary Fig. 1b). Fig. 2a showed pathological sections of 12 pairs of AH and TH patients selected for single-cell sequencing, excluding the presence of

malignant lesions, and follicle size increases with the increase of AH and TH grades (please see Supplementary Fig. 2b).

ii) As for Fig. 3a and 4a, immunohistochemistry of B cells and T cells was performed using CD20 and CD3 antibodies, respectively, indicating a significant infiltration of B cells and T cells in AH and TH. And through quantitative results, we found that the proportion of B cells in AH was higher than that in TH, while the proportion of T cells was lower than that in TH (please see Supplementary Fig. 4a and 6a).

- Figure 2C: The order of column labels is inverted.

Response:

Sorry for our negligence. When integrating this picture, we accidentally mirrored them. The picture has now been corrected and redrawn from scale RNA expression to average RNA expression (please see Fig. 2c) in the revised manuscript.

- Figure 3B: The shape of the UMAP is the same as for the total cells (Figure 2B). The B cells should be re-plotted as in Figure 4B for T/NK cells.

Response:

Thanks for the reviewer's constructive suggestion. We have redrawn the UMAP plots in Fig. 3b and 4b in the revised manuscript.

- Figures 3D-3E: The pseudotime appears biased by the cell cycle, as naive and memory cells (mostly quiescent) are at the beginning of pseudotime, whereas they typically appear later. Mention or correct for these cell cycle biases in the text. This also applies to T cells in Figure 4E (lines 188-191).

Response:

Thanks for reviewer's reminder. Our data have already undergone cell cycle correction during preprocessing in the original manuscript. The proportion of G1, G2M, and S phase cells in each B

cell and T cell subtypes was supplemented in Supplementary Fig. 4f and Supplementary Fig. 6g. Furthermore, by changing the gene expression value from scale average RNA expression to average RNA expression and adding the expression of *DUSP4*, *FCRL5*, *ZEB2*, *ITGAX*, and *FCRL4*, we corrected memory B cells to atypical memory B cells^{2,3}. It had been reported that naïve B cells and immature atypical memory B cells were located in tertiary lymphoid structures and developed into plasma B cells through canonical germinal center and alternative extrinsic pathways, respectively². Therefore, we believed that the B cell differentiation trajectory in the manuscript was consistent with the development process of B cells. Similarly, the differentiation trajectory of T cells also followed the developmental process of naïve T cells-cytotoxic T cells-exhausted T cells⁴.

- Figure 4B: The colors in the UMAP do not correspond to the legend. The light blue CD4 intermediate cluster is missing, while there are two separate yellow clusters for CD4 inhibitory.

Response:

Thanks for reviewer's reminder. We have redrawn the UMAP plots of T cell subtypes (please see Fig. 4b).

- Comparing Figures 4B and Supplementary Figure 4D: There are doubts about the cell type assignment. Correct the cluster definition as CD8 is expressed in the CD4 naïve cluster.

Response:

Thanks for reviewer's reminder. We have redrawn the gene expression UMAP plots of T cell subtypes based on the average RNA expression of marker genes (please Supplementary Fig. 6d) in the revised manuscript.

- Lines 195-197: Since the difference is not significant, the authors cannot claim there are more intermediate CD4 T cells.

Response:

Thanks for the reviewer's reminder. We have deleted this part of the content in the revised

manuscript.

- Lines 199-217: The GO analysis misses the column bar information and must be normalized to expressed genes to avoid biases in data interpretation. The whole paragraph is mostly speculative without data validation.

Response:

This section contained the analysis results of GSEA (Gene Set Variation Analysis). Based on the reviewer's suggestion, we have normalized the RNA expression of input genes using the "scale" function in base R software package (version 4.3.0) and recalculated the GSEA score of the pathway. Moreover, GSEA is a particular type of gene set enrichment method and enables pathway-centric analyses of molecular data by performing a conceptually simple but powerful change in the functional unit of analysis, from genes to gene sets⁵. In addition, we have used flow cytometry to validate the expression of core genes in differential pathways, further clarifying the proportion and functional differences of naïve CD4⁺ T cells and regulatory CD4⁺ T cells between AH and TH (please see Pages 12-13, lines 239-257).

Minor Comments

1. The unit is missing from the scale bar in Figures 1F, 2C, 2D, 3C, 4C, 5A, 5C, 5G, 5H, S1A-B-C-D-H, S3B, S4D, and S4E.

Response:

Sorry for our negligence. We have added all missing units in the figures in the revised manuscript.

2. Line 89: "4-5 years" should be changed to "2-5 years."

Response:

We have corrected this error (please see Page 5, lines 83-84) in the revised manuscript.

3. In the figures, "celltypes" should be corrected to "cell types."

Response:

Thanks for the reviewer's constructive suggestion. We have revised Fig. 2e, Fig. 3f, Fig. 4f, Supplementary Fig. 3d-g, Supplementary Fig. 5a-d, Supplementary Fig. 7b-f in the revised manuscript.

4. Figure 1E: The blue triangle for the years' scale bar is unnecessary, as is the vertical triangle for grade level since it's not a gradient of grades but rather two discrete grades.

Response:

Thanks for the reviewer's constructive suggestion. We have removed the blue triangle (please see Fig. 1e) in the revised manuscript.

7. Figure 2: Remove titles from the figures.

Response:

Thanks for the reviewer's constructive suggestion. We have removed the titles from Fig. 2 in the revised manuscript.

8. Figures 2B, 2C, 2D, and XX: Replace "Plasma" with "Plasma B Cells" as the former is misleading.

Response:

We have revised "Plasma" to "Plasma B cells" in Fig. 2b-e, Fig. 3b-d, Fig. 3f,g, Fig. 7a,b, Fig. 7d-e, Supplementary Fig. 2a, Supplementary Fig. 2i,j, Supplementary Fig. 3d-g, Supplementary Fig. 4d-f, Supplementary Fig. 5a-d, Supplementary Fig. 8a,b in the revised manuscript.

10. Figure 2D: The scale bar is missing, and the fluorescence intensity should be quantified.

Response:

Sorry for our negligence. We have added scale bar and fluorescence quantification results (please see Fig. 2d and Supplementary Fig. 3c) in the revised manuscript.

11. Figure 2E: The color code of patients doesn't seem to match the dots. It might be better to use green dots for AH, yellow for TH, and use the patient's color code for the matching lines. The name of the cells analyzed in each plot should be reported above the plots.

Response:

Thanks for the reviewer's constructive suggestion. We have revised Fig. 2e in the revised manuscript.

14. Figure 3D: AH and TH should be labeled at the top of the plots.

Response:

Sorry for our negligence. We have revised Fig. 3d in the revised manuscript.

15. Figure 4: Better define the categories of CD4-intermediate in the text, as they are not a biologically distinct category.

Response:

Thank you for reviewer's suggestion. We have renamed CD4-Intermediate and CD4-Inhibitory to CD4-Inhibitory1 and CD4-Inhibitory2, and described CD4-Inhibitory1 as exhausted effector memory CD4⁺T cells (please see Page 10, lines 204-205). Moreover, the differences between CD4-Inhibitory1 and CD4-Inhibitory2 have been described in detail in lines 206-210 and lines 221-224 in the revised manuscript.

18. Supplementary Figure 4D: Better define NK cells markers.

Response:

Thank you for reviewer's suggestion. We have modified the gene *NCAMI* as an NK cell marker and

redrawn the result (please see Supplementary Fig. 6d) in the revised manuscript.

19. Figure 5A: The percentage cannot exceed 100%.

Response:

Sorry for our negligence. The y-axis represented the number of cells, which exceeded 100. We have redrawn Fig. 5a using cell percentages. In the new version of the manuscript, it has been labeled as Fig. 6a.

20. Figures 5G-H: Description of the columns is missing, and the figure legend says “left and right” instead of “top and bottom.” Without this information, it is difficult to understand the results.

Response:

We selected some meaningful pathways in Fig. 5g and showed the expression of core genes of these pathways in Fig. 5h. We have added textual descriptions in Fig. 5g and 5h to enhance readability. In the new version of the manuscript, they were Fig. 6g,h.

21. Figures 4G-H and 5G-H: Significant changes in p-values should be reported to support the conclusions of the data analysis.

Response:

Fig. 4g-h and 5g showed the analysis results of GSVA (gene set variation analysis). The GSVA package provides the implementation of four single-sample gene set enrichment methods, concretely zscore, plage, ssGSEA and its own called GSVA. We have normalized the RNA expression of input genes using the “scale” function in base R software package (version 4.3.0) and used the “gsva” analysis method, which was a non-parametric method that uses the empirical CDFs of gene expression ranks inside and outside the gene set, but it started by calculating an expression-level statistic that brings gene expression profiles with different dynamic ranges to a common scale. Therefore, the GSVA score of the gene sets obtained by the “gsva” method was a calibrated and

tested value, with positive values assigned to positively regulated pathways and negative values assigned to negatively regulated pathways^{6,7}. So we directly selected the pathways based on the positive or negative values of the GSVA score. The *P* value of Fig. 5g was calculated using the "FindMarkers" function of the Seurat R package and has been added to Fig. 5h. In the new version of the manuscript, they were Fig. 5a,b and Fig. 6g,h.

22. Figure 6A: This analysis would be more meaningful on differentially expressed genes between AH and TH. The same applies to Figures B and C.

Response:

Fig. 6a-c showed the cell-cell communication and differential expressed ligand-receptor pairs. The analysis methods for these results were based on CellphoneDB⁸, a database of ligand-receptor genes, to examine the expression of ligand receptor genes in cell types. The focus of the results was to highlight the expression differences of ligand-receptor genes, which were not related to the differentially expressed genes between AH and TH. We first calculated the cell communication situation and then selected differential ligand-receptor genes for display.

23. Figure 6D: Data must be normalized as percentages since different patients have different numbers of sequenced cells.

Response:

Thanks for the reviewer's constructive suggestion. We have revised Fig. 6d-e, 6g, Supplementary Fig. 5b and Supplementary Fig. 5d. In the new version of the manuscript, they were Fig. 7d-e, 7g, Supplementary Fig. 8b and Supplementary Fig. 8d.

24. Figures 6F-H: Quantify the immunofluorescent images to draw a conclusion.

Response:

We have added the quantitative results of dual fluorescence co-localization in Supplementary Fig.

8d in the revised manuscript.

25. Figure S2E: The scale bar is between -1 and 0. Is this correct? What is the unit of average expression?

Response:

Thanks for the reviewer's reminder. Gene expression values were normalized, resulting in a narrowed range of gene expression, or even negative values. We have redrawn this figure using average RNA expression values in the revised manuscript (please see Supplementary Fig. 2i).

26. Figures S3B/S4A: There are too many patients for one plot. Either make one plot per patient or remove this plot (as the information is somewhat included in Figure S2A).

Response:

Thanks for the reviewer's constructive suggestion. We have deleted Supplementary Fig. 3b and 4a in the revised manuscript.

27. Line 103: "children s" should be corrected.

Response:

We have corrected this error (please see Page 5, line 94) in the revised manuscript.

28. Line 104: Explain what "the ratio" means in the text. Additionally, there is no panel G in Figure 1.

Response:

The collection and calculation methods of peripheral blood indexes have been supplemented on pages 19-20, lines 410-417. "The ratio" refers to the proportion of cell type in the total number of white blood cells in 1L peripheral blood. Moreover, we have corrected the error in the image numbering in the revised manuscript.

29. Lines 105-106: Clarify the concept of “being important to infection through immunological responses.” The same applies to lines 132-133.

Response:

We have revised these conclusions (please see Page 6, lines 106-109 and Pages 7-8, lines 146-147) in the revised manuscript.

30. Lines 108-109: Rephrase these lines. The aim of the project is to describe the transcriptional status during disease.

Response:

We have revised the description of this section (please see Page 6, lines 111-113) in the revised manuscript.

31. Lines 124-125: Rephrase these lines. The percentages likely refer to “the total number of cells in each tissue” rather than “the total number of B cells.”

Response:

We have revised the description of this section (please see Page 7, lines 137-138) in the revised manuscript.

32. Line 169: Highlight the significance of the difference between AH and TH T/NK cells in the figure.

Response:

We have already modified the content of this section (please see Page 10, lines 196-198) in the revised manuscript.

33. Line 175: Better discuss the distinction between intermediate and proliferating CD4 cells.

Response:

We did not believe that there was a significant correlation between CD4-Intermediate and CD4-Proliferation, but CD4-Intermediate and CD4-Inhibitory also exhibit significant exhausted states. We have renamed CD4-Intermediate and CD4-Inhibitory to CD4-Inhibitory1 and CD4-Inhibitory2, and described CD4-Inhibitory1 as exhausted effector memory CD4⁺T cells (please see Page 10, lines 204-205). Moreover, the differences between CD4-Inhibitory1 and CD4-Inhibitory2 have been described in detail in lines 206-210 and lines 221-224 in the revised manuscript.

35. Line 193: Modify "The that of regulatory..." to "the percentage of..."

Response:

We have corrected this error (please see Page 11, line 226) in the revised manuscript.

37. Line 240: "Figure 5G" should read "Figure 5g" for consistency.

Response:

We have corrected this error (please see Page 15, line 305-306) in the revised manuscript.

38. Lines 71-72: These conclusions seem to contrast with lines 311-313. Rephrase these lines for clarity.

Response:

We have removed the content in lines 311-313 from the original manuscript.

References:

1. Liu A, *et al.* A rat model of adenoid hypertrophy constructed by using ovalbumin and lipopolysaccharides to induce allergy, chronic inflammation, and chronic intermittent hypoxia. *Animal Models and Experimental Medicine*, (2024).
2. Ma J, *et al.* A blueprint for tumor-infiltrating B cells across human cancers. *Science* **384**, (2024).

3. Hao Y, O'Neill P, Naradikian MS, Scholz JL, Cancro MP. A B-cell subset uniquely responsive to innate stimuli accumulates in aged mice. *Blood* **118**, 1294-1304 (2011).
4. Kim N, *et al.* Single-cell RNA sequencing demonstrates the molecular and cellular reprogramming of metastatic lung adenocarcinoma. *Nat Commun* **11**, 2285 (2020).
5. Hanzelmann S, Castelo R, Guinney J. GSEA: gene set variation analysis for microarray and RNA-seq data. *BMC Bioinformatics* **14**, 7 (2013).
6. Luo Y, *et al.* Single-cell transcriptomic analysis reveals disparate effector differentiation pathways in human T(reg) compartment. *Nat Commun* **12**, 3913 (2021).
7. Costa B, *et al.* Human cytomegalovirus exploits STING signaling and counteracts IFN/ISG induction to facilitate infection of dendritic cells. *Nat Commun* **15**, 1745 (2024).
8. Garcia-Alonso L, *et al.* Single-cell roadmap of human gonadal development. *Nature* **607**, 540-547 (2022).

Comments from Reviewer 1

Remarks to the Author:

The authors have successfully addressed all reviewer's comments.

Response:

We thank the reviewer for reviewing our manuscript and endorsing our work.

Reviewer #2 (Remarks to the Author):

Following peer review, the authors revised the findings and added functional analysis.

Additional functional analysis includes rat data. However, some of the answers are scientifically questionable and the methods used by the authors might not be appropriate. In particular, it has been anatomically and immunologically pointed out that rats and other rodents do not have tissue equivalent to the human tonsils, so the use of rats is not appropriate. Additionally, batch consolidation in single-cell analysis may be inappropriate.

Response:

Thanks for the reviewer's constructive suggestion. We have modified the purpose of adding a rat model experiment in the revised manuscript (please see Pages 8, lines 160-162). Meanwhile, we added separate analysis of scRNA-seq data for AH and TH, and found that the comparison of the proportion of cell types analyzed separately was consistent with the results of integrated analysis in the revised manuscript (please see Page 7, lines 132-138, Page 8, lines 155-156, Supplementary Fig. 3a-e).

My several concerns about the methodology and data analysis are below.

1) The references used as the definition of hypertrophy in adenoid and tonsils are not appropriate.

Response:

Thanks for the reviewer's constructive suggestion. We have revised the references on the grade of AH and TH (please see Pages 4-5, lines 79-81).

Comments: In the authors' revised version, the TH grade has been removed from the table, which is no longer informative. The grade of TH should be added to the vertical axis of Table 1 to show how much of each grade of TH is present in each AH and whether there is a correlation between the grade of AH and TH.

Response:

Thanks for the reviewer's reminder. We have added statistics on TH grading in Tables 1 and 2. Interestingly, as the grade of AH increased, the proportion of patients with TH grade IV decreased (please see Table 1), which was worth further in-depth research.

2) It would be informative to show the number of people, sex ratio, age, and BMI for each of AH+TH+ and AH+TH-. Different grading systems are also used for the tonsils and adenoids, so each stage should be listed.

Response:

Thanks for the reviewer's constructive comments. We redefined AH grade I as AH-, and showed the number, sex ratio, age, and BMI of patients with AH grade I-IV in Table1. The number, sex ratio, age, and BMI of patients in AH-TH-, AH+TH-, AH-TH+, AH+TH+ groups are shown in Table2.

Comments: Table 2 shows that the range of BMI for each group was very wide. Therefore, it is necessary to show the percentage of underweight or overweight children based on BMI. The normal range for BMI in children varies with age and sex. The definition of overweight or underweight showed at CDC homepage (<https://www.cdc.gov/bmi/child-teen-calculator/bmi-categories.html>).

In addition to that the accompanying diseases of OSAS shown in lines 53-54 of the introduction are reports of adults, and OSAS in children has been reported to be accompanied by growth disorders (low BMI) and restlessness etc., which is consistent with the information provided by the authors especially in the AH+TH+ group. It would be informative that lines 53-54 of the introduction need to be revised according to the information in table 2 (The references should be changed to include comorbidities in pediatric OSAS).

Response:

Thanks for the reviewer's constructive suggestion. We have added statistics on BMI category in Tables 1 and 2. The definition of BMI category was developed by the National Center for Health Statistics in collaboration with the National Center for Chronic Disease Prevention and Health

Promotion (2000). <http://www.cdc.gov/growthcharts>. Three patients under the age of 2 were not included in the statistics. The statistical results of BMI category in Table 1 supported our conclusion that BMI of patients were negatively correlated with the AH grade (please see Page 5, lines 88-91). In addition, we have revised the description of the correlation between pediatric OSAS and weight in the **Introduction** section in the revised manuscript, and modified the references on the pathological changes of AH combined with TH (please see Page 3, lines 52-56).

6) In Fig. 1 and sup Fig. 1, there is no mention of when and how the composition of blood immune cells was collected and analyzed. Is it blood cell measurement, automatic measurement, single cell, or flow cytometry, and if it was taken before surgery, were there any changes after surgery? The authors conclude that the difference in the composition of blood lymphocytes is the result of a local immune response in the upper respiratory tract. Still, it is challenging to conclude without comparing the results of blood tests before and after surgery.

Response:

The peripheral blood indexes were obtained by performing complete blood counts on the patients during the consultation, which were obtained through Flow Cytometry method with semiconductor laser on Automated hematology analyzer Sysmex (please see Pages 19-20, lines 410-417). We counted the indexes of immune cells. At present, postoperative recovery is not based on changes in peripheral blood indexes, so patients would not receive complete blood counts during follow-up visits, which were in line with diagnostic and treatment standards. Therefore, we have added the AH-TH- group as a healthy control and revised the conclusion of this section (please see Pages 5-6, lines 95-103, Table 2 and Supplementary Fig. 1d-g).

Comments: In the end, there is no clear statement as to when the blood was taken except for grade I patients who did not undergo surgery, whether before or after the surgery, and I believe that my questions are not answered yet.

Response:

We apologize for not providing a clear explanation in our initial response. The patients included in our cohort analysis were children who visited our hospital's outpatient clinics for evaluation due to suspected AH. From this group, we selected those who underwent complete blood counts, nasal

endoscopy, and pressure plate examinations during their medical visits, totaling 1,209 cases. Not all of these patients underwent surgery at our hospital. Additionally, even for those who did undergo surgery, complete blood counts were not performed postoperatively. Consequently, we are unable to assess changes in peripheral blood parameters before and after surgery. Our analysis of peripheral blood parameters was primarily aimed at demonstrating the differences in peripheral blood immune cell composition among patients with different AH grades and between patients with AH with or without TH.

7) If the authors would like to compare AH and TH, the tonsils and adenoids should be analyzed separately in Fig. 2, single-cell analysis.

Response:

Adenoids and tonsils are both gland tissues that play an immune function, and their cell composition is similar. The integration analysis of 24 sample data from 12 patients could help remove the batch effect, better normalize the expression level of genes, and help to identify the cell types and gene expression. At present, the R software package Seurat also recommends the integration analysis of multiple tissues and samples, and has launched a variety of methods to integrate gene expression. Our data use the FindIntegration Anchors function and IntegrateData function recommended by the software package to integrate the sample data, which could better present the cell type and gene expression of each sample¹. Moreover, Supplementary Fig. 2h shows the UMAP map of cell types of AH and TH, respectively. Supplementary Fig. 2j shows the marker gene expression of cell types from AH and TH, respectively.

10) Analyzing and showing AH+ and TH+ single-cell data separately in all figures would be informative.

Response:

Thanks for the reviewer's constructive suggestion. The UMAP maps of the cellular distribution of AH and TH were shown in Supplementary Fig. 2h, Supplementary Fig. 4b and Supplementary Fig. 6b. Moreover, we have added the marker gene expression of main cell types, B cell subtypes and T cell subtypes of AH and TH (please see Supplementary Fig. 2j, Supplementary Fig. 4e and Supplementary Fig. 6f) in the revised manuscript.

Comments: The methods section states that the FindIntegration Anchors function and IntegrateData function were used for individual datasets (not between tissues).

Furthermore, based on the method, it can be interpreted that the authors performed single-cell analysis by grinding adenoid and tonsil tissues together from the beginning (before applying the samples BD Rhapsody). Batch consolidation is typically performed based on data from independent applying, especially for tissues such as adenoids where single cell analysis cell characterization of tissues has not yet been established (bioRxiv, PMID: 39484391).

Again, please state in the method whether the adenoids and tonsils were mixed to create a cell suspension before application. If the analysis uses a premixed cell suspension, the methodology is scientifically questionable. If not, analyze the samples separately and then combine the data to compare. Analysis and consideration is required before and after the integration.

Response:

We apologize for not providing a clear explanation in the previous version of the manuscript. We collected single-cell suspensions of 24 samples separately for subsequent library construction, without mixing them together. When constructing the libraries, we used BD™ Human Single-Cell Multiplexing Kit (BD, 633781). This kit reduces costs by assigning special labels to each sample, allowing for the mixing of multiple samples for single-cell capture. In subsequent analysis, the data of a single sample can still be obtained by de-multiplexed. The quality of the single sample data obtained by this method is consistent with that of the data from a single sample without mixing. We have added these descriptions about library construction methods in the **Methods** section in the revised manuscript (please see Page 21, line 434 and lines 441-444).

For scRNA-seq data analysis, we corrected the expression matrices of 24 samples separately and integrated 12 AH samples and 12 TH samples into one AH dataset and one TH dataset using FindVariableFeatures, FindIntegrationAnchors and IntegrateData function. Although FindIntegrationAnchors and IntegrateData function were commonly used for integration between multiple samples of the same tissue, Hoa Thi Nhu Tran et al. found that by evaluating Local inverse Simpson's index integration (iLISI) scores, FindIntegrationAnchors and IntegrateData function exhibited better performance of batch mixing when integrating scRNA-seq data from multiple

human organs¹. Although FindIntegrationAnchors and IntegrateData function was not the optimal method for integrating scRNA-seq data from two tissues, considering the similar tissue sources and cell compositions of adenoids and tonsils, we used FindIntegrationAnchors and IntegrateData function to integrate the AH dataset and TH dataset for subsequent comparative analysis. We have added these contents to the **Methods** section in the revised manuscript (please see Page 22, lines 463-470).

We have added the results of separate analysis of AH dataset and TH dataset, and found that both AH and TH contained B cells, T and NK cells, plasma B cells, myeloid cells, plasma cell-like dendritic cells, epithelial cells and neutrophils, and the results of cell proportion analysis were consistent with those of integrated analysis (please see Page 7, lines 132-138 and Page 8, lines 153-156). Therefore, we believed that the integrated datasets of AH and TH could be used for comparative analysis of the two tissues.

13) Most of the data is based on single-cell analysis, and to verify this, functional analysis of sorted B cells and T cells may be necessary. For example, analysis of antibody induction ability and cytokine production ability against stimulation by foreign antigens, LTA, and LPS.

Response:

Thanks for the reviewer's constructive suggestion. We have added functional analysis of B and T cells in AH and TH, and collected 6 pairs of additional AH and TH samples, including 3 cases of AH grade III, 3 cases of AH grade IV, and 9 cases of TH grade II (Supplementary Table 3). And we have also collected peripheral blood from children who came to the hospital for physical examination as a healthy control. Using antibodies CD20 and CD3 respectively, B cells and T cells were sorted from tissues and peripheral blood by flow cytometry. The results showed that compared to TH, AH contained a higher proportion of B cells and fewer T cells (Fig. 5e). After 36 hours of in vitro culture, cytokine detection was performed on the cell supernatant. Through evaluating the levels of cytokines that perform different functions, we determined the main functions of B and T cells in AH and TH in the revised manuscript. Specifically, compared to TH, T cells in AH had more significant immune suppression, chemotaxis, and apoptosis, while B cells had weaker antigen presentation and proliferation abilities. Notably, the stimulation of LPS amplified the difference in secretion function

between two cell types in two tissues (please see Pages 13-14, lines 262-284).

Comments: Please show the gating strategy for each cell sorting. Without that gating data, it is impossible to determine whether the method is scientifically valid.

It has been suggested that immune cells in AH are more exhausted than those in TH, but there is insufficient discussion of this finding (Differences in the functions, cellular composition, characteristic feature and epithelial characterization of the two tissues etc..).

Response:

Thanks for the reviewer's reminder. We have added gating strategies for B cell and T cell sorting in the **Methods** section in the revised manuscript (please see Page 28, lines 602-608). Specifically, A) Exclude doublets from single cells through the area and height of FSC. B) Using FSC and SSC parameters to remove debris from single cells and selecting complete clusters of target cells. C) Remove dead cells from the target cells by labeling with BD Pharmingen™ 7-AAD (BD, 559925). D) Select all white blood cells using CD45 antibody in live cells, and E) further select T cells and B cells using T cell specific antibody CD3 and B cell specific antibody CD20, respectively. F) Then, sort these two cell populations, as showed in the figure below:

Major Comments

Comments: The authors added rat experiments based on a report by Anqi Liu et al. Unfortunately, their report examined hypertrophy of the NALT. Tonsillar tissue does not exist in rats, mice, or hamsters. It remains to be seen whether rodent NALT is an anatomically and immunologically (PMID: 21869895 Casteleyn C, et al., 2011, Clin Dev Immunol).

Response:

Thanks for the reviewer's reminder. The comment is very correct, as reviewer mentioned, tonsillar tissue does not exist in rats, mice, or hamsters. It remains to be seen whether rodent NALT is an anatomically and immunologically. Therefore, we have modified the purpose of adding a rat model experiment in the revised manuscript (please see Page 8, lines 160-162). Based on the experimental results in this section, we believe that the activated type of immune cells during AH and TH are commonly present in the inflammatory response of the nasopharynx.

Comments: Throughout the manuscript, the terms adenoid hypertrophy and adenoid (or

tonsil hypertrophy and tonsil) are used interchangeably, which can be confusing. Specifically, AH and TH in Line 429 and Line 430 should be described as adenoid (tissue) and tonsil (tissue), respectively. It is important to note that AH and TH are conditions and not tissue or the entities themselves. The same goes for Line 523, 539.

Response:

Thanks for the reviewer's reminder. We carefully checked and corrected these errors (please see Page 21, lines 440-441, Page 26, lines 563 and Page 27, lines 579).

Reference

1. Tran HTN, *et al.* A benchmark of batch-effect correction methods for single-cell RNA sequencing data. *Genome Biol* **21**, 12 (2020).

Reviewer #3 (Remarks to the Author):

While this reviewer appreciates the improvements made to the manuscript and figures, the response to some of my initial comments remains inadequate and misleading.

For instance, (lines 258-284) the authors have included peripheral blood (PB) from healthy controls to strengthen their conclusions, but they have compared PB cells from healthy controls to AH and TH cells from patients. What it is presented as of now, is not a straightforward comparison between healthy and diseased conditions, but rather a comparison between different tissue sources. It is well established that immune cells vary significantly depending on the tissue in which they reside.

Response:

Thanks for the reviewer's reminder. The comment is very correct, as reviewer mentioned, PB cells from healthy controls, AH and TH belong to different tissue sources and cannot be directly compared. We have removed the direct comparison results of cytokine secretion between PB and AH/TH in the revised manuscript (please see Fig. 5f,g). The data from PB was only used to compare the effects of LPS treatment (please see Fig. 5f,g and Supplementary Fig. 9a,b)

Moreover, the authors did not normalize the data. At the very least, they should have set the control values in Figure 5F to “1” to emphasize the differences between pre- and post-LPS treatment. The authors state that “after LPS treatment... cytokine secretion of B cells and T cells in AH and TH was more vigorous.” However, normalization reveals a different picture. For instance, IL-8 increases approximately fivefold in controls (from ~1 to ~5), whereas the increase in AH and TH is only between 1.2 and 2 times. This reviewer would conclude the opposite of the authors' statement. The same applies to G-CSF, MIP-1a, MIP-1b and maybe to others.

Response:

Thanks for the reviewer's constructive suggestion. We normalized the data of PB, AH, and TH

before LPS treatment to 1 and re-evaluated the effect of LPS treatment on cytokine secretion. We found that compared with the PB group, LPS did not significantly increase the secretion of IL-8, MIP-1 α , and MIP-1 β , and IL-6 in B and T cells of AH and TH, which may be related to the already severe inflammatory response of B and T cells in AH and TH (Supplementary Fig. 9a,b). In addition, LPS had inhibitory effects on the secretion of MIF, SCGF- β and IL-16 in PB, AH and TH (Supplementary Fig. 9a,b). We have added this section to Page 14, lines 280-284 (Supplementary Fig. 9a,b).

Under this reviewer's suggestion, the authors corrected the pseudotime of B cells, since memory B cells were strangely at the beginning of pseudotime. However, it is not so clear how they perform this normalization, they have just shown the percentages of cells in the different cell cycle stages. The authors affirmed they "correct by changing the gene expression value and by \div some gene expression". What does it mean to "add" some gene expression? Overall, this reviewer does not think they solved this issue. The authors have just renamed the clusters of memory B cells under the name of "atypical memory B cells", but they did not detect clusters of memory B cells in the plot (Figure 3E). To our knowledge atypical memory cells should co-exist with memory cells, not substitute for them. The general pattern of differentiation of B cells is not clear from their data. The authors should try to explain this phenomenon or remove this analysis.

Response:

In the preprocessing of scRNA-seq data, we first extracted marker genes from the S and G2M phases of cells, sorted and scored B cell subtypes using the CellCycleScoring function, and corrected the expression values based on the difference between the S and G2M scores for subsequent pseudo-time analysis.

The meaning of "add" referred to adding the result of the expression of CD20, DUSP4, FCRL5, ZEB2, ITGAX, and FCRL4 in B cell subtypes. Based on this result, we renamed memory B as atypical memory B. We did not believe that the two types of cells were interchangeable, but we did not identify memory B (no cells expressed CD27) in our study, which may be limited by sample preparation and sequencing techniques. Due to the limited types of identified B cell subpopulations,

we have removed the results of B cell subtypes differentiation. We have identified the co-expression of MKI67 and IGHG1 in proliferating B cells and determined the presence of highly proliferating plasma B cells in TH (please see Page 9, lines 184-185).

Moreover in Fig4E all CD4/CD8/NK cells have been analyzed in the same plot. There is no meaning in putting different cells from different differentiation processed in the same pseudotime. As CD4 won't convert into a CD8 cell (lines 217-220 must be corrected).

Response:

Thanks for the reviewer's reminder. We have redrawn the differentiation trajectory of CD4⁺ T cell subtypes (please see Page 11, lines 227-230).

Most of the enriched GSEA terms in CD4-Naive and Treg (Figure 5A-B) are generic and do not underline significant functions/roles of these cells between AH and TH tissues. The conclusions drawn from that plot are lengthy somewhat meaningless.

Response:

In order to obtain the GSEA pathway specifically expressed in naïve CD4⁺ T cells and regulatory CD4⁺ T cells, we used FindAllMarkers function to identify marker genes for naïve CD4⁺ T cells and regulatory CD4⁺ T cells. If the differential pathway contained at least two marker genes, the pathway would be selected and showed. We have added this section to Pages 23-24, lines 495-499. Moreover, we have also reduced the description of this result in the main text (please see Pages 12-13, lines 250-257).

Minor comments

Fig 4D panel of TNF, the scale bar is different in the upper facs plot with respect to the bottom. Moreover the facs plot suggests a bimodal distribution, not quantified in the MFI.

Response:

Sorry for our negligence. We did not select the correct gating strategy when drawing the TNF

histogram of TH, resulting in incorrect results. We have already modified this figure in the revised manuscript (please see Fig. 5d).

“Lefse” at lane 125 must be explained

The metagenomics analysis is missing from the Material and Methods Section.

Response:

Linear discriminant analysis (LDA) effect size (LEfSe) was used to identify the microbial taxa that significantly differed between different groups. It used nonparametric tests to identify significant features, performed subclass comparisons, and then LDA to estimate the effect size of identified features. We have added explanations for LEfSe and methods for constructing and analyzing of mNGS in the revised manuscript (please see Page 7, lines 129 and Pages 24-25, lines 519-536).

Fig s2e the enrichment seems to be opposite with the data presented in Fgi S2c/d. For instance Bacteroidetes is enriched in AH in Fig S2C, but the enrichment is in TH in Fig S2e (and so on).

Response:

Sorry for our negligence. When we redrawn the figure of this result, we assigned the wrong colors to each group in the previous version of the manuscript. We have already modified the figure in the revised manuscript (please see Supplementary Fig. 2f). The original image is as follows:

Lines 230-235 can be summarized for clarity

Response:

We have revised the content of this section in the revised manuscript (please see Pages 12-13, lines 250-257).

Reviewer #4 (Remarks to the Author):

Response:

We thank the reviewer for reviewing our manuscript.

Comments from Reviewer 2

Remarks to the Author:

Although this reviewer appreciates the improvements made to the manuscript and figures, this reviewer believes that the overall conception of the paper is unclear.

Specifically, is the theme of this study about the "hypertrophy pathogenesis" or "the differences in immune responses between adenoids and tonsils"?

Response:

Thanks for the reviewer's reminder. We have described the theme of our study as “the differences in immune responses of adenoids and tonsils after hypertrophy by infection” (please see Page 6, lines 107, Page 12, lines 241, Page 8, lines 155, Page 14, lines 282, and Pages 17-18, lines 361-362).

According to the paper PMID: 38301653 (cited by the authors) and the preprint paper PMID: 39484391, the cellular composition of the adenoid and tonsil is markedly different. In particular, the characteristics of epithelial cells are notably distinct. However, this study barely mentions the cluster of epithelial cells.

There have been some concerns about the methodology of single-cell analysis in the past, and this reviewer has confirmed the method during previous peer reviews. Upon further review, this reviewer found that the authors' study did not demonstrate entirely significant differences between the cell clusters of the adenoid (AH) and the tonsil (TH).

Response:

Ramon Massoni Badosa et al. generated an atlas of the human tonsil composed of > 556000 cells, across five different data modalities, including single-cell transcriptome, epigenome, proteome, and immune repertoire sequencing, as well as spatial transcriptomics (PMID:38301653). Their study used not only children's tonsils, but also young and old people's tonsils. Notably, these tonsils were removed not only because of recurrent tonsillitis, but also in the operation of benign pharyngeal squamous papillomatosis. Meanwhile, Samuel Alvarez-Arguedas et al. collected from 6 donors (age 2-14; 3 males and 3 females) undergoing elective adenoidectomy for obstructive sleep apnea, and

performed single-nucleus RNA sequencing on these samples (PMID:38301653). The two studies found that the cell composition of adenoids and tonsils were significantly different because of their different sample ages, sample preparation methods, and sequencing methods.

However, in our study, 1) the samples for scRNA-seq were all from children aged 2-5 years old. 2) All children were diagnosed with AH combined with TH, so we collected adenoids and matching tonsils from the same child. 3) We ensured that all adenoid or tonsillar tissues were cut in similar locations during sampling and the same sample preparation methods were used. 4) For the scRNA-seq data of 12 tissues, we used the same quality control and analysis methods. In summary, the cell composition of AH and TH in our study was similar, with only differences in the number and function of some immune cell subtypes. These experimental design and results could better demonstrate the theme of our study, which was the differences in immune responses of adenoids and tonsils after hypertrophy by infection.

Due to the limitations of our sample preparation method, we did not collect abundant epithelial cells, which we have described in the **Discussion** section (please see Page 19, lines 401-404).

Since epithelial cells are considered extremely important as triggers of immune responses, it seems complicated to overlook this cluster. It is necessary to discuss the differences from previous reports thoroughly.

The strength of single-cell analysis is that it is possible to analyze rare cell groups, so the small number of cells is not a limitation.

Response:

Thanks for the reviewer's constructive suggestion. We have added the analysis of cell communication between epithelial cells and B cell subtypes, T cell subtypes to evaluate the situation of epithelial cells triggering immune response. The results showed that compared to TH, increases cell communication between the epithelial cells and naïve CD4+T cells, regulatory CD4+ T cells, exhausted effector memory CD4+ T cells, exhausted CD4+ T cells, cytotoxic CD8+ T cells and DNT in AH (Supplementary Fig. 11a), which might be due to the role played by ligand-receptor pairs LGALS9/CD45 (Supplementary Fig. 11b). These results indicate the presence of a unique immune regulatory pattern of epithelial cell activated T cell subtypes in AH. (please see Page 17,

lines 347-353). And we have described in the **Discussion** section (please see Page 17, lines 359-360).

Reviewer #3 (Remarks to the Author):

The authors have addressed all my comments.

Response:

We thank the reviewer for reviewing our manuscript and endorsing our work.

Reviewer #4 (Remarks to the Author):

Response:

We thank the reviewer for reviewing our manuscript and endorsing our work.